# Simultaneously ultrafast and robust two-dimensional flash memory devices based on phase-engineered edge contacts

Jun Yu[1], Han Wang[1], Fuwei Zhuge [1]✉, Zirui Chen[2], Man Hu[1], Xiang Xu[1], Yuhui He[2], Ying Ma [1]✉, Xiangshui Miao [2] & Tianyou Zhai [1]✉

As the prevailing non-volatile memory (NVM), flash memory offers mass data storage at high integration density and low cost. However, due to the 'speed-retention-endurance' dilemma, their typical speed is limited to ~microseconds to milliseconds for program and erase operations, restricting their application in scenarios with high-speed data throughput. Here, by adopting metallic 1T-$Li_xMoS_2$ as edge contact, we show that ultrafast (10–100 ns) and robust (endurance>$10^6$ cycles, retention>10 years) memory operation can be simultaneously achieved in a two-dimensional van der Waals heterostructure flash memory with 2H-$MoS_2$ as semiconductor channel. We attribute the superior performance to the gate tunable Schottky barrier at the edge contact, which can facilitate hot carrier injection to the semiconductor channel and subsequent tunneling when compared to a conventional top contact with high density of defects at the metal interface. Our results suggest that contact engineering can become a strategy to further improve the performance of 2D flash memory devices and meet the increasing demands of high speed and reliable data storage.

In the era of digital technology, massive data awaiting processing and storing keeps calling for high-density, ultrafast, and robust memory technology[1–3]. In the last decade, as the elemental device in solid state disk, flash memory had evolved markedly to fulfill the requirement of high-volume density by moving from planar configuration to 3D packing[4,5]. However, for decades, the operation speed for the prevailing flash memory products in market barely improved due to the dilemma of 'speed-retention-endurance' in optimizing their performance. Increasing the operation speed of flash memory, e.g., by large operation voltage, typically sacrifices its endurance and retention[6]. This is because even though charge injection efficiency for program/erase could be theoretically improved by high electric field, the cell tends to fail early due to the accelerated defect generation and growth in tunneling dielectric[7,8]. Unintentional interfacial roughness in device also focuses local electric field and adversely degrades cell lifetime[9].

Though channel or source side hot carrier injection had been successfully exploited to enhance the charge injection efficiency without using high voltage, fast operation speed ~sub-microsecond is only achieved for program, and the erase speed is still slow ~milliseconds[10,11].

Ideal Schottky contacts had been predicted with the ability to inject hot carriers to semiconductor channel under gate bias modulation, and hold the potential to dramatically boost the performance of flash memory[12]. However, in conventional Si technology, achieving an abrupt Schottky contact is challenging due to issues related to inter-diffusion and Fermi level pinning (FLP)[13–15]. With their atomically flat layer structure, the emerging two-dimensional van der Waals (2D vdW) materials have numerous ways to engineer an ideal Schottky barrier, including using phase-engineered or vdW contact[16,17]. They could also be integrated at high packing density yet with low defect density and

[1]State Key Laboratory of Materials Processing and Die and Mould Technology, School of Material Science and Engineering, Huazhong University of Science and Technology, Wuhan 430074, China. [2]Hubei Yangtze Memory Laboratory; School of Integrated circuits, Huazhong University of Science and Technology, Wuhan 430074, China. ✉e-mail: zhugefw@hust.edu.cn; yingma@hust.edu.cn; zhaity@hust.edu.cn

clean interfaces[18,19], which are desirable for ultrafast and robust memory operation by providing an ideal charge injection interface and durable tunneling layer[20]. However, the performance of 2D flash memory in previous literatures falls behind of the expectation in spite of various channel and float gate used[21–27]. Until recently, 20–160 ns superior operation speed was achieved in InSe and MoS₂ flash memories based on the clean interface in vdW heterostructures or hot carrier injection directly though the ultrathin 2D material[28,29]. However, a competing endurance lifetime to the prevailing Si flash technology (>10⁵ cycles)[30,31] was rarely demonstrated for a simultaneously ultrafast and robust flash memory. To this end, a recent example that adopts bipolar WSe₂ as the channel displays the potential in achieving well-balance memory performances by combine the Lucky-electron injection mechanism, with an oxide charge trapping layer structure for endurance enhancement[32]. On the other hand, though an ideal Schottky contact could theoretically enhance charge injection efficiency in a flash memory, the vital role of contact interface in leveraging memory performance had been overlooked in the past, especially considering that the ultrathin two-dimensional crystal lattice is extremely sensitive to direct metal deposition in the conventional top contact configuration.

Here, we demonstrate the realization of simultaneously ultrafast (program/erase speed ~10/100 ns) and super-robust (endurance lifetime >10⁶ cycles) based on phase-engineered edge contacts to 2D MoS₂ flash memory. If compared to traditional top metal contacts that have rich lattice and electronic defects due to metal-induced gap states (MIGS) or interdiffusion impurities[16], lateral edge contacts exhibit highly tunable Schottky barrier and render efficient hot carrier injection into the semiconductor channel during program/erase (P/E) operation. This markedly improves the charge injection efficiency to float gate and guarantees ultrafast and super-robust memory operation at the same time, which was been rarely reported in 2D flash memory. The comprehensively optimized key figure of merits over the prevailing commercial flash memory makes our edge-contacted 2D flash memory a viable option for high-speed and durable memory in the future.

## Results

### Ultrafast P/E speed in memory cells with edge contact

Figure 1a illustrates the structure of our flash memory made of a vdW stacking of MoS₂/hBN/few-layer graphene (FLG) on top of a thermal oxidized Si substrate. The phase-engineered edge contact is made by

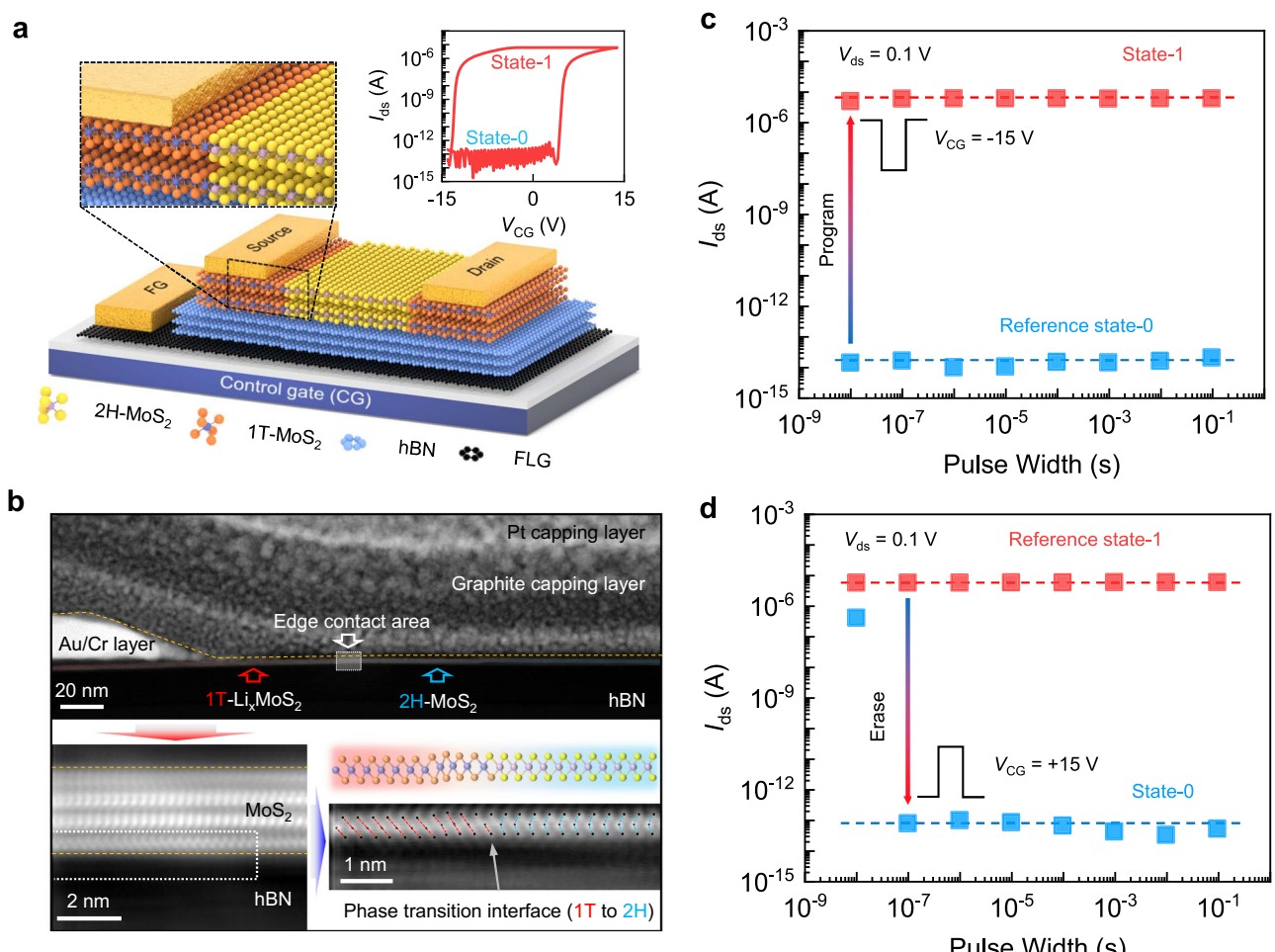

**Fig. 1 | Ultrafast MoS₂ flash memory with phase-engineered edge contact.**
**a** Illustration of the flash memory based on vdW heterostructure of MoS₂/hBN/few-layer graphene (FLG), the edge contact is formed by patterned 1T-Li$_x$MoS₂ under conventional Cr metal contact. The inset hysteresis loop shows clearly the switch of readout current ($I_{ds}$) between state-0 and state-1 via sweeping the control gate bias ($V_{CG}$). **b** Atomic-scale observation of the edge contact in MoS₂, the high-angle annular dark field scanning transmission electron microscopy (HAADF-STEM) image validates the phase transformation of 2H-MoS₂ into a distorted 1T phase. The atomic arrangement of S-Mo-S in the bottom layer is marked using red and blue dash lines to highlight the 1T and 2H phase transition interface. **c**, **d** reveal the program and erase performance when varying the width (10 ns to 100 ms) for applied $V_{CG}$ pulse (−15 V for program, and 15 V for erase). The reference states (state-1 in **c** and state-0 in **d**) were set by initializing voltage pulses with 100 ms width. Red and blue dash lines are guidelines that indicate the level of ON and OFF states by a successful program/erase operation.

patterned lithium intercalation in n-butyl lithium solution[33,34], which transforms the semiconductor MoS$_2$ (2H) layers into metallic phase (1T-Li$_x$MoS$_2$) (see method and Supplementary Note 1 for fabrication details). Cr/Au contact is followingly made for electrical connection to the transformed metallic phase. The typical thickness of MoS$_2$ and hBN in the prepared memory cells is 3–5 nm and 11–12 nm (Supplementary Note 2). Using high-angle annular dark field scanning transmission electron microscopy (HAADF-STEM) image (Fig. 1b), the vdW interfaces and edge contact via phase transition in the fabricated heterostructure are confirmed. A transition from 2H-MoS$_2$ in channel to a distorted 1T-MoS$_2$ structure could be identified near the contact pad, forming the lateral edge contact. Due to the interlayer diffusion of lithium, the 1T-2H interface extrudes slightly from the lithography patterned area in to the channel, and has slight structure distortion in plane due to intercalation-induced strain. In our design, the phase transformation in MoS$_2$ is ensured to reach the bottom layer from the top surface, which avoids intercalation at the hetero-interface, e.g., the one between MoS$_2$ and hBN. The extinction of photoluminescence spectra of 2H-MoS$_2$ is used in experiment to verify the degree of phase transformation (Supplementary Note 3). When sweeping the voltage bias applied to control gate (CG), the device displays an apparent memory window in the inset of Fig. 1a, which stem from charge trapping in floating gate (FG) rather than hBN or SiO$_2$ dielectric layer (Supplementary Note 4).

Using ultrafast electric pulses applied to control gate, we successfully program/erase the above memory cell within 10/100 ns (pulse waveform discussed in Supplementary Note 5). As indicated in Fig. 1c, when starting from a reference OFF state (state-0, erased by $V_{CG} = 15$ V for 100 ms), the memory cell is fully programmed into state-1 by 10 ns pulse at $V_{CG} = −15$ V, reaching a high ON/OFF ratio ~10$^7$. At a lower voltage of ~7V, the memory was still successfully programmed with an on/off ratio>10$^4$. The significantly lower operation voltage than previous reports suggested high charge injection efficiency in present memory cell, which is vital for later robust endurance behavior. Reversely, when starting from a reference ON state (state-1, programmed by $V_{CG} = −15$ V for 100 ms), the memory cell is fully erased into state-0 within 100 ns ($V_{CG} = 15$ V). If compared to the prevailing silicon flash, the above ultrafast P/E speed is markedly improved by 2–4 orders respectively. According to the tunneling barrier for electrons ($\Phi_{tB}^e = 2.8$ eV, for erase) and holes ($\Phi_{tB}^h = 2.0$ eV, for program) from MoS$_2$ conduction band (CB) and valence band (VB) through hBN (Supplementary Note 6), such ultrafast P/E operation speed could theoretically be achieved considering a strong electric field in hBN ($E_{hBN} > 10$ MVcm$^{-1}$) during the applied voltage pulse (Supplementary Note 7), which is met in our device based on its high gate coupling ratio ~0.9 (Supplementary Note 8). However, it is worth noting that the high operation voltage itself does not guarantee high P/E speed in practical devices, and the edge contact by 1T-Li$_x$MoS$_2$ is considered as the other key factor in improving the charge injection efficiency for ultrafast operation.

To validate the critical role of edge contact in enhancing the operation speed, we fabricated paired devices on the same MoS$_2$/hBN/graphene vdW heterostructure using edge contact (1T) and conventional top face contact (Cr) respectively, as indicated in Fig. 2a and b. The paired devices have identical thickness for each constituting layers (MoS$_2$, 4.2 nm; hBN, 15.8 nm; Gr, 2.8 nm, Supplementary Note 9) and gate coupling ratio (GCR), therefore comparable electrical field strength in tunneling hBN layer. In Fig. 2c, d, their P/E performance is compared directly by varying both the amplitude and width of applied P/E pulses. For each measurement, the memory was initially set to a saturated reference state-0 or state-1 using 100 ms electric pulses. Despite of additional etching process in fabricating paired memory cells, the obtained edge-contacted memory cell have identical P/E characteristic to the one from direct vdW stacking (Supplementary Note 10). If taking an on/off ratio of 10$^4$ as the criteria for successful P/E

operation, the edge contacted cell has 2–3 orders faster P/E speed than top contacted memory cell at the same operation voltage, which clearly indicated the superior charge injection efficiency offered by edge contact. In the laboratory, we have fabricated and evaluated more than 49 memory cells (26 cells with 1 T edge, and 23 cells with Cr top contact) to compare their P/E performances. The thickness of hBN layer in these memory cells distributed within the range of 8–17 nm with an average value of ~11 and ~12 nm for top and edge contacted memory cells (Supplementary Note 2). In Fig. 2e, f, the P/E speed of all memory cells is summarized by the attained ON/OFF ratio (relative to the reference state) after applying 10/100 ns P/E pulses of varying amplitudes. Despite the device-to-device (D2D) variation by fabrication, statistically, we could find that replacing Cr contact using 1T-Li$_x$MoS$_2$ markedly improves the yield of high-speed memory cells. According to the cumulative analysis, >90% memory cells with edge contact can be successfully programmed (defined as when an acceptable ON/OFF ratio ≥10$^2$ is attained) within 10 ns at the normalized operation voltage of 14 V for 10 nm hBN layer ($V_{norm} = V_{pulse}/t_{hBN} \times 10$ nm, where $t_{hBN}$ represents the thickness of hBN layer in device). Comparatively, only ~40% is achieved for top-contacted memory cells. For erase, the statistical yield at 100 ns is also improved from 20% to 40%. The lower rate for erase is related to the large tunneling barrier through hBN for erase, and may be improved via designing the band alignment in vdW heterostructure for a more balanced P/E performance.

## Edge contact facilitated charge tunneling injection to float gate

Hot carrier injection had been known occur through Schottky barrier in metal contacted field-effect transistors, and had been exploited to boost the program speed of float gate memory to microseconds[11,35]. Though almost all 2D float gate transistors are made with Schottky contacts, fast operation speed was not always guaranteed[21–27]. In Fig. 3a, b, we illustrate the tunneling pathways for edge and top contacted memory cells and the related potential barrier for hole tunneling, which is responsible for program in our memory (Supplementary Note 6). Previously, hole tunneling had been discussed responsible for the operation of 2D float gate transistors that use MoS$_2$ as the channel and graphene as float gate[36]. In our case, the dramatic enhancement of P/E speed via contact engineering to MoS$_2$ suggest that electron/hole tunneling from MoS$_2$ to graphene is most likely responsible for the memory operation. Here, hole tunneling is used for illustration considering the apparent improvement to program speed if compared to erase. Although MoS$_2$ displays n-type conduction with electrons as the majority carriers, the high field modulation under positive gate bias during program is sufficient to induce significant amount of holes at the level of 10$^{13}$ cm$^{-2}$ if considering a field strength of 10 MV/cm across hBN.

Recent studies on MoS$_2$ memory cells have shown that vertical charge tunneling from metal contact to bottom 2D semiconductor is critical to generate hot carriers for ultrafast memory operation[29]. However, in our case, the measured tunneling current density via 1T-MoS$_2$/hBN/Gr pathway is ~2 order lower than that through 2H-MoS$_2$ due to the higher tunneling barrier from the Fermi level of 1T-MoS$_2$ (Supplementary Note 11). By comparing directly the tunneling current across edge contacted 2H-MoS$_2$/hBN/graphene heterostructure, we found ~2 orders enhancement to conventional top contacted one at the electric field of 8 MV/cm (Supplementary Note 11). This suggest that the edge contact interface plays a critical role in enhancing the charge injection efficiency. In addition, replacing graphene with MoS$_2$, a symmetric tunneling structure of MoS$_2$/hBN/ MoS$_2$ with respective edge and top contact to channel and float gate displays highly asymmetric P/E speed, showing an ultrafast program within 10 ns but slow erase >100 ms (Supplementary Note 12). This again confirms the adoption of edge contact to MoS$_2$ channel contributes to the observed difference in charge injection efficiency. Considering the edge contact

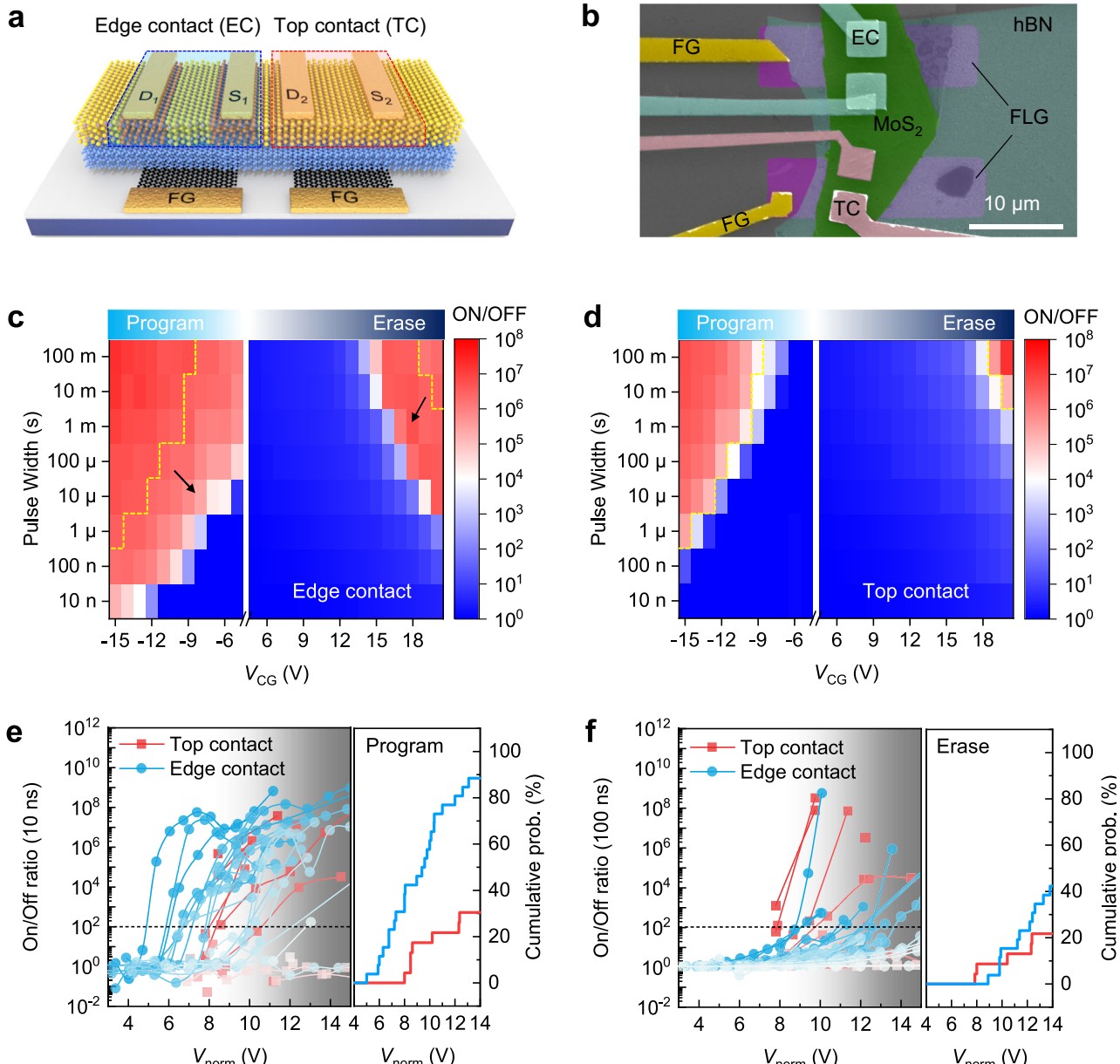

**Fig. 2 | Comparison of the program/erase (P/E) performance of memory cells with 1 T edge or Cr top contacts. a, b** Schematic illustration (**a**) and false color scanning electron microscopy (SEM) image (**b**) of the paired memory cells on the same vdW heterostructure. The float gate (FG) is made by etching an exfoliated few-layer graphene. Thus, the paired memory cells exhibit identical thickness combination for each layer and differ only in contact configuration using 1 T edge (EC) or Cr top contact (TC). **c, d** Map of the attained ON/OFF ratio of memory under different voltage pulse conditions when changing both the amplitude and pulse width: 1 T edge contact (**c**) and Cr top contact (**d**). The P/E condition that guarantees a high ON/OFF ratio = $10^4$ in top contacted memory cell is marked in both figures for a guidline (dash line) when comparing their operation speed. **e, f** P/E behavior (**e, f**) of >20 memory cells under ultrafast electric pulse (10 ns for program, 100 ns for erase) and varying operation voltage. The operation voltage was normalized according to the thickness of tunneling hBN to reflect the electric field strength at the tunneling layer. On the right panel of **e, f**, the cumulative probability is counted if an ON/OFF ratio ≥$10^2$ is attained by the applied ultrafast P/E pulses.

configuraiton, we expect that instead of tunneling from contact region, the charge injection in edge contacted memory cell tends to initiate via a lateral pathway, which crosses the phase change interface from 1 T to 2H-MoS₂. In this case, the band bending ($\Phi_s$) in 2H-MoS₂ accelerates the injected carriers from Schottky contact and efficiently lowers the tunneling barrier through hBN layer via hot carrier effect ($\Phi_{tB}^h = \Phi_{tB0}^h - \Phi_s$). In comparison, for the case of conventional top contact, the Fermi level in MoS₂ tends to be pinned by the trap states under contact affected (CA) area, which reduces $\Phi_s$ under gate modulation[16,37]. By transforming the CA area in to a metallic phase, the present 1 T contact extends from the defined top contact area by

lithium diffusion in interlayer space, thus making the charge injection via lateral Schottky junction immune to manufacturing defects.

The low trap density in edge contacted memory cell is confirmed by the frequency dependence of the capacitance-voltage (C-V) characteristic (Fig. 3c), which is measured between the MoS₂ channel and float gate. Within the frequency range of 4–100 kHz, the measured capacitance ($C_p$) for edge contacted memory cell displays slight frequency dependence, while $C_p$ for Cr top contacted memory cell reduces apparently with increasing the frequency, especially when $V_{FG}$ is above the flat band voltage ($V_{FB}$). Using the high-low frequency method, the interface trap density was determined according to

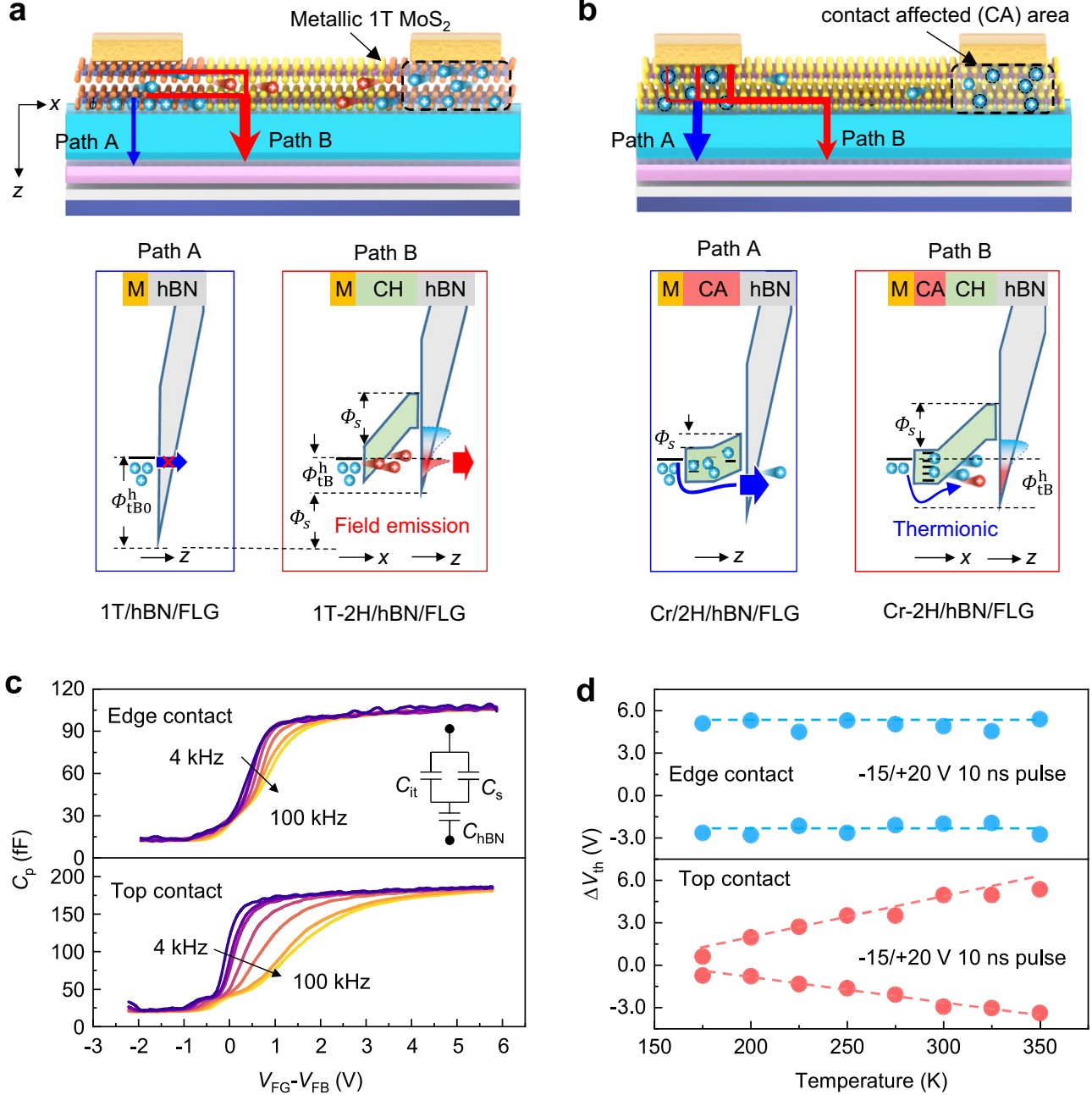

**Fig. 3 | The tunneling pathway and charge injection behavior for edge or top contacted flash memory.** Illustration of the energy band diagram for charge injection during program operation in **a** edge and **b** top contacted memory cells, the tunneling pathways through contact region and semiconductor channel are compared (M: metal contact, CH: channel, CA: contact affected area). When considering the band bending in semiconductor channel ($\Phi_s$) under field modulation, the hole tunneling barrier ($\Phi_{tB}^h$) though the valance band of hBN is apparently reduced from the initial value ($\Phi_{tB0}^h$). If compared to the field emission (indicated by red arrow) in edge contact, rich trap states under contact affected area (CA) in top contact configuration lead to thermionic process governed charge emission at contact (blue arrow). **c** Capacitance-voltage characteristic and **d** temperature-dependent charge injection behavior of edge and top contacted memory cell. The measured capacitance is intepreated using the inset equialivent circuit, which considers capacitance associated with interface states ($C_{it}$), semiconductor channel ($C_s$), and hBN dielectric layer ($C_{hBN}$). **d** The charge injection efficiency is reflected by the threshold shift under ultrafast (10 ns) P/E operation, in which the pulse amplitude are respectively −15 V and 20 V for program and erase. Their distinct temperature dependence is indicated by the dash guidelines.

$qD_{it} = \left[ \left( \frac{1}{C_{LF}} - \frac{1}{C_{hBN}} \right)^{-1} - \left( \frac{1}{C_{HF}} - \frac{1}{C_{hBN}} \right)^{-1} \right]$, in which $C_{LF}$ and $C_{HF}$ are respectively the areal capacitance at low and high-frequency limit, and $q$ the elementary charge[38]. For top and edge contacted memory cell, the integrated trap density was ~$2.2 \times 10^{13}$ cm$^{-2}$ and $8 \times 10^{11}$ cm$^{-2}$ (Supplementary Note 13), respectively. The high trap density in conventional top contacted memory cells may come from both electronic

traps generated from electron-beam lithography and also lattice disorders caused by depositing top metal contact (Supplementary Note 14). Since the response time of these trap states lag significantly behind of the applied 10–100 ns pulse duration for ultrafast P/E operation, they act as fixed space charges that increase the potential barrier for charge injection[39]. The additional capacitance related to interfacial trap states also tend to reduce the potential drop in MoS$_2$,

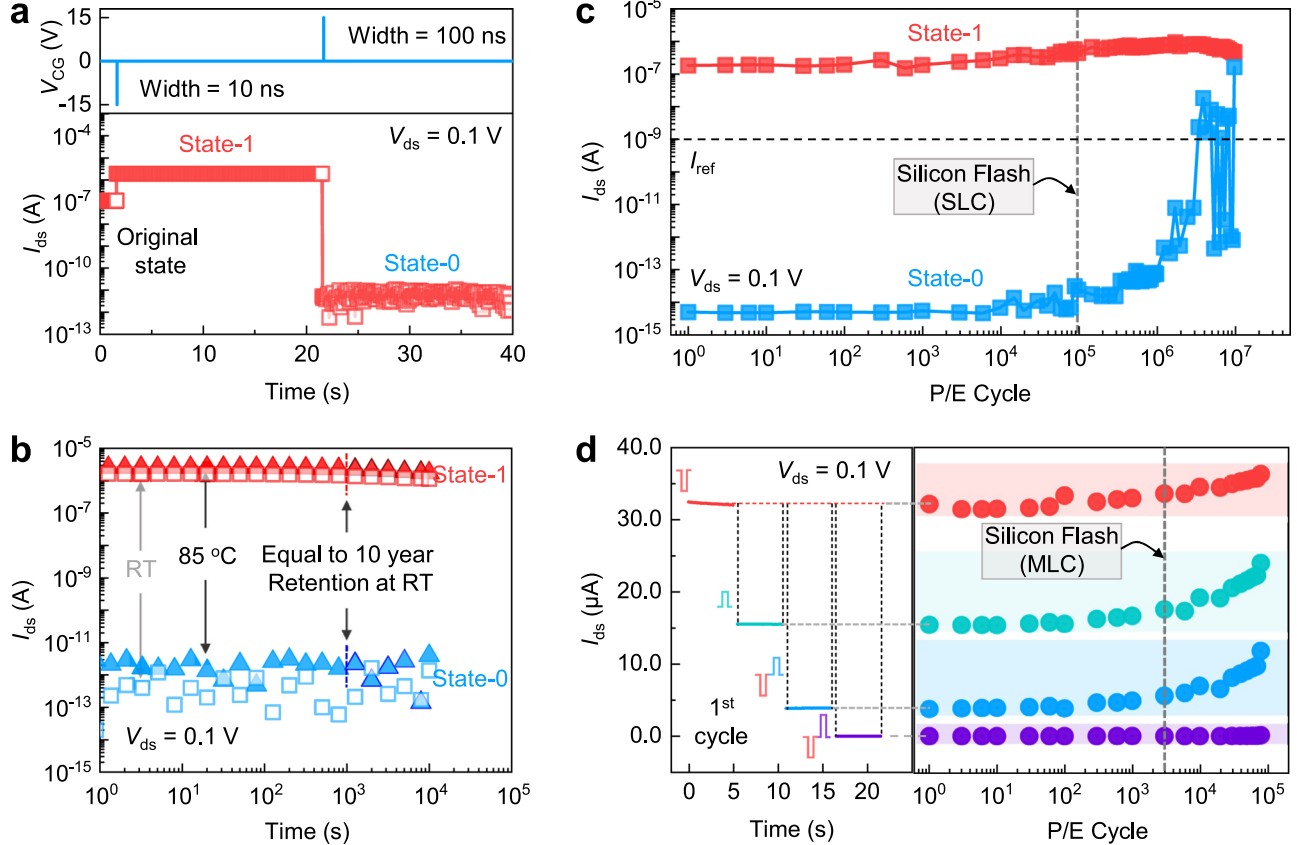

**Fig. 4 | Ultrafast operation speed, long-term retention, and robust endurance characteristics simultaneously attained in edge-contacted MoS₂ flash memory.** **a** Monitored change of memory state during sequentially applied ultrafast P/E pulses. **b** Retention performance of the memory under RT (open square), and 85 °C (filled triangle) acceleration, with an acceleration ratio of ~3.3 × 10⁵ at 85 °C, 10-year data retention is expected at the vertical dash line. **c** Endurance characteristic of the memory as single-level cell (SLC). The programming and erasing operations are performed by (−8 V, 10 ns) and (15 V, 100 ns) operation pulses, respectively. Memory failure is confirmed if the readout at erase state become higher than the defined $I_{ref} = 1$ nA, indicated as horitontal dash line. **d** Endurance characteristic of the memory as multi-level cell (MLC). Four memory states marked in shaded area is achieved during the repeated P/E operation. As SLC and MLC, the endurance lifetime is ~3 × 10⁶ cycles and 8 × 10⁴ cycles respectively. **c, d** Vertical dash lines that indicate the commercial standard of endurance lifetime for Silicon flash (10⁵ cycles as SLC, and 10³ cycles as MLC) are included for comparison.

thus counteracts the barrier lowering for hot carrier injection. In present float gate memory cells, the reduced trap density in edge contact renders highly tunable Schottky barrier under gate modulation (Supplementary Note 15), which is essential for hot carrier injection via Schottky emission at the metal-semiconductor interface. Consistently, when the edge contact is placed away from float gate coupling (Supplementary Note 9), the absence of Schottky barrier modulation at the metal contact interface during memory operation deteriorates apparently the charge injection efficiency by edge contact. This suggests that gate coupling to edge contact interface is crucial for achieving superior charge injection efficiency.

The essential role of trap effects in impeding ultrafast memory operation is further supported by the distinct temperature effect to memory operation in the paired memory cells. Figure 3d displays the extracted threshold voltage shift ($\Delta V_{th}$, read at $I_{ds} = 20$ μA with $V_{ds} = 1$ V) of memory at the temperature range from 100 to 300 K after ultrafast P/E operation. By fixing the P/E pulses (P: −15 V, 10 ns; E: 15 V, 100 ns), $\Delta V_{th}$ reflects directly the charge injection efficiency in memory. For edge contacted memory cell, $\Delta V_{th}$ is found independent of $T$, which agrees with the behavior of FN tunneling limited charge injection that is weakly influenced by $T$[40]. After a short program pulse, the measured $\Delta V_{th}$ reaches as high as 7–9 V for all temperature, corresponding to a high charge density of ~5.7 × 10¹¹ cm⁻² in float gate. In comparison, the paired Cr contacted memory cell displayed slow operation speed and apparent $T$ dependence of $\Delta V_{th}$, which suggested

a thermal activation behavior for charge injection. An Arrhenius fitting to the result yields shallow activation energy of 58 meV and 40 meV for trapped charges during program and erase operation respectively[36].

**Robust endurance of edge contacted flash memory**

The significantly improved charge injection efficiency in the above via edge contact essentially allows us to realize ultrafast P/E performance without sacrificing retention or endurance characteristics. Figure 4a, b displays respectively the ultrafast P/E performance and long data retention characteristics of an edge-contacted memory cell. The memory states are rapidly switched with a superior ON/OFF ratio >10⁷ via 10 ns program (−15 V) and 100 ns erase (15 V) pulses. Notably, the obtained state-0 and state-1 in above memory display retention >10⁴ s at both room temperature (RT) and 85 °C acceleration (Fig. 4b). According to a leakage barrier of ~1.95 eV and related acceleration ratio (AR) of 3.3 × 10⁵ (Supplementary Note 16), this value suggests an equivalent data retention lifetime >10 years (~3.3 × 10⁸ s) at RT, which stems from the excellent charge trapping ability of graphene and low defect states in hBN[22,28,29].

In Fig. 4c, the endurance behavior of an edge-contacted memory cell as a single-level cell (SLC) is revealed. The memory state is switched between state-0 and state-1 by P/E pulses of (−8 V, 10 ns) and (15 V, 100 ns). Notably, if compared to previously reported ultrafast InSe (20 V, 20 ns) and MoS₂ (30 V, 20 ns) flash memory cells that adopted top metal contacts[28,29], our memory is operated at

considerably lower voltage based on its superior charge injection efficiency. This endows the memory cell an exceeding endurance lifetime ~$3 \times 10^6$ cycles (Fig. 4c). The barely drifted readout current for state-0 and state-1 before $10^6$ cycles suggest well-suppressed stress in hBN layer. As discussed in Supplementary Note 17, the memory after $10^5$ endurance cycles still keeps long data retention over years. Comparatively, top contacted memory cells working at the same operation voltage require longer pulse duration to reach the same on/off ratio, and results in typical endurance lifetime ~$10^4$ cycles (Supplementary Note 18). According to the analysis of optical images of failed devices and time-to-breakdown of the heterostructure (Supplementary Note 18), we associate the superior endurance lifetime in edge-contacted memory cells to the suppressed interfacial roughness, which in top-contacted memory cells may be introduced from the lattice distortion in $MoS_2$ by direct metal evaporation. The present edge contact strategy based on in-situ phase transition transforms 2H-$MoS_2$ under metal contact into its metallic 1 T phase, as an interfacial layer, it avoids undesired roughness at contact interface. With an endurance lifetime >$10^6$ cycles, the edge-contacted memory cell meets the requirement of the prevailing silicon flash memory (~$10^5$ cycles for SLC)[41-44], while having 2–4 orders' faster P/E speed (Supplementary Note 19).

Finally, we show the potential of edge-contacted memory cells as multi-level cell (MLC) to increase the bit density. It is worth noting that though multiple memory states (e.g., 4-bits) can be written in memory by altering the pulse condition (Supplementary Note 20), the aggressively narrowed margin among states is prone to deteriorate the attained endurance lifetime due to the sensitive drift of stored states by stress effect in hBN layer. Though hBN is exfoliated as a single crystal, it still exhibits a certain level of trap states that could store charges during repeated P/E cycles. Under the high electric field, more traps will be created by the defect generation, which in turn exacerbates the drift of memory states[45]. Thus, we instead choose to realize an MLC with 2-bit storage, with state-0 written via a program pulse (−8 V/10 ns for state-0), and states-1, 2, and 3 by erase pulses of varying amplitudes (8, 10, 12 V/100 ns) following the initial state-0. As indicated in Fig. 4d, the stored states are well separated in linear space, and could still be clearly distinguished after $8 \times 10^4$ cycled P/E operation. Such performance is ~1 order better than the commercial standard (~$3 \times 10^3$ cycles for MLC) while having the ultrafast P/E speed[44]. In the future, further study on the integration of such memory cells in NAND or NOR architectures could promote its application in high-density mass-storage memory[46]. Meanwhile, for high-density integration of dramatically scaled memory cells with reduced channel and contact size, precise control of the position and quality of phase-engineered contact would be critical. As mass-storage memory, such dramatically improved P/E speed to <100 ns and multi-bit performance is favored by scenarios that demand high throughput data processing and storage, i.e., cloud servers, video recording, and analysis, etc.

## Discussion

In summary, by introducing phase-engineered edge contact, we have demonstrated the realization of simultaneously ultrafast P/E speed ~10/100 ns, stable data retention >10 years, and super-robust endurance lifetime (>$3 \times 10^6$ cycles for SLC, and $8 \times 10^4$ for MLC) in $MoS_2$ flash memory. We associate our device performance to the lateral hot carrier injection by edge contact under gate modulation, enabling a markedly lower P/E voltage for robust endurance lifetime while maintaining a decent charge injection efficiency for high-speed operation. The comprehensively improved key figures of merit over the existing commercial flash memory devices indicate a potential strategy to break the 'speed-retention-endurance' dilemma in conventional charge-based memory, making our 1 T contacted memory cell a viable option for high-speed and robust flash memory in the future.

## Methods

### Fabrication of 2D flash memory and characterization

The vdW heterostructures were made by sequentially stacking graphene, hBN, and $MoS_2$ obtained via mechanical exfoliation on a 300 nm $SiO_2$/Si substrate using PDMS-assisted dry transfer processes. The thickness of each layer was characterized via atomic force microscopy (Dimension Icon, Bruker). First, few-layer graphene (FLG) nanosheets were transferred onto the highly doped p-type silicon substrate that was cleaned by three-minute oxygen plasma in advance. After that, hBN and $MoS_2$ nanosheets were transferred to the top of FLG successively by dry transfer method with the help of PDMS (polydimethyl siloxane). As PDMS residues were sticky, an annealing process was carried out to remove residues and obtain a clean interface. Next, PMMA (polymethyl methacrylate) was spin-coated onto the fabricated heterostructures, and patterned by electron-beam lithography. Cr/Au (10 nm/40 nm) electrodes were then deposited by thermal evaporation. For edge-contacted devices, the heterostructures were immersed in 3 ml of n-butyl lithium (n-BuLi, Aladdin) in a sealed container at room temperature in a glovebox (ref. 33). Lithium intercalation into the interface between $MoS_2$ and hBN layer should be avoided. After soaking in the n-BuLi for 2.5 h, the samples were washed with hexane to remove excess n-BuLi. The sample was evaluated by Raman and photoluminescence (Alpha 300 R, WITec) to confirm the phase transition before depositing metal contact. For the fabrication of paired memory cells, FLG nanosheet transferred onto substrates was etched by Ar plasma to create two separate FLG flakes with identical shape and area.

### Electrical characterization of memory

The electrical testing of the memory devices was performed at room temperature and in vacuum (if not specified) in the probe station (TTPX, Lake Shore). A semiconductor characterization system (B1500A, Keysight) is used to perform the electrical measurements. The direct current signals were generated using the source/monitor unit in the B1500A, while ultrafast electric pulses were generated using a Pulse/Pattern Generator (81110 A, Agilent). The waveform of generated pulse was recorded with an oscilloscope with a bandwidth of 2.5 GHz (DSO9254A, Keysight). All the electrical testing was performed in a vacuum condition to avoid the effect of the atmosphere.

## Data availability

The Source Data underlying the figures of this study are available at https://doi.org/10.6084/m9.figshare.23994789. All raw data generated during the current study are available from the corresponding authors upon request.

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

## Acknowledgements

This work was supported by the National Natural Science Foundation of China under Grant No. 21825103 (T.Z.) and No. U21A2069 (T.Z.), Item "large-scale and energy-efficient in-memory computing systems" of National Key Research and Development Program of China (Y.H.), and Ministry of Science and Technology of China under the grant No. 2021YFA1200500 (Y.M.). We also thank the technical support from the Analytical and Testing Center at Huazhong University of Science and Technology.

## Author contributions

F.Z. and T.Z. conceived and supervised the project. J.Y. and H.W. contributed equally to this work, by performing most of the device fabrication and electrical characterization. M.H., X.X. participated partly in device fabrication. Z.C., Y.H., and X.S.M. provided support to the simulation and endurance measurements. F.Z., J.Y., and H.W. conducted data analysis by discussing with all authors. F.Z., J.Y., H.W., Y.M., and T.Z. co-wrote the manuscript and revised it by discussion with all authors.

## Competing interests

The authors declare no competing interests.
