## [Peer Review File · Nature Communications]

Simultaneously ultrafast and robust two-dimensional flash memory devices based on phase engineered edge contactsEditorial Note:

This manuscript has been previously reviewed at another journal that is not operating a transparent peer review scheme. This document only contains reviewer comments and rebuttal letters for versions considered at *Nature Communications*.

Parts of this Peer Review File have been redacted as indicated to remove third-party material where no permission to publish could be obtained.

Reviewer: 1

In this manuscript by Yu et al., the authors used the metallic 1T-LixMoS₂ as edge contact in flash devices for achieving non-volatile super-robust memory. The authors made the paired devices on the same exfoliated MoS₂/hBN/graphene stack using 1T-contact and conventional top-contact with a 2H-MoS₂ channel. The non-volatile memory operation is both ultrafast in the paired devices but the 1T contact cell is more robust than the common top-contact one. Thus, the authors claimed that the 1T-MoS₂ phase-engineered edge contact is useful for robust memory technology. This manuscript suffers from several serious problems concerning the significant novelty insights, mechanism clarification and experimental design in this work, and it is not suitable for a high-standard and impactful journal like Nature Electronics.

The novelty of this work is doubtful:

Q1:

The biggest problem with the manuscript is that there is no first-hand novelty in the concept of using two-dimensional semiconductors as the channel material to realize ultrafast flash memory. The authors claimed that the edge-contact enable a super-robust (endurance $>10^6$) ultrafast flash memory. Two papers by Wu et al. (Nat. Nanotechnol. 16, 882–887; 2021) and Liu et al. (Nat. Nanotechnol. 16, 874–881; 2021) independently reported an ultrafast programming speed non-volatile 2D flash memory device. An endurance of 8×10^6 cycles for ultrafast 2D flash (Nat. Nanotechnol. 18, 1–7; 2023) was also reported by the same team mentioned above. All of the above works have used the conventional top-contact and the endurance of these devices can also larger than 10^6 . Therefore, we can not agree with the authors that the edge-contact has any performance enhancement of the endurance.

Reply:

We thank the reviewer's comments. We'd like state that achieving the enhancement in speed and endurance simultaneously is a highly challenging task due to the dilemma of "speed-endurance trade-off". In 2021, two reports achieved ultrafast speed at 20 ns level based on the vdW heterostructured flash memory. [1, 2] However, a competing endurance lifetime at that moment is not demonstrated. By using a new contact engineering strategy based on in-situ phase change, our work achieved simultaneously ultrafast memory operation within 10-100 ns while having a competing endurance lifetime that meets the standard of commercial silicon flash.

We note that our strategy is different with the mentioned publication in Nat. Nanotechnol. 18, 1–7; 2023, [3] which is published during the review process of our manuscript. The report exploits Lucky-electron tunneling and a charge trapping stack of hBN, HfO₂ and Al₂O₃ to improve speed and endurance, respectively. Our strategy based on contact engineering had not been reported at the moment.

We further note that the superior memory performance from edge contacted memory cell is validated from both paired memory cells, and also statistical comparison on more than 40 memory cells (**Figure 2e and f**). The critical role of edge contact in enhancing operation speed is also

supported by our additional experiment, in which we replace graphene float gate using MoS₂, thus forming a symmetric heterostructure of edge-MoS₂/hBN/MoS₂-top. It is apparent in the following **Figure S6-5** that the device exhibits highly asymmetric P/E speed. The memory cell can be programmed within 100 ns, but can be only erased with >100 ms pulses. Such highly asymmetric P/E behavior in the symmetric heterostructure with only contact difference indicated the critical role of contact mode. The result can be properly understood using hot hole injection via the highly tunable edge contact, since the injection via top contacted 2H-MoS₂ is significantly impeded by the access region from metal contact. Compared with the program operation speed (100 ns), the erase operation speed (>100 ms) is slower.

Figure S6-5. Edge contacted flash memory with MoS₂ as the float gate. **a**, Configuration schematic diagram of the memory device. **b**, Energy band diagram of the memory under program (left)/erase (right) operation. **c** and **d** reveal the program and erase performance when varying the width for applied V_{CG} pulse (-14 V for program, and 14 V for erase). The reference states (state-1 in **a** and state-0 in **b**) were set by initializing voltage pulses with 100 ms width.

Reference:

1. Wu, L. et al. Atomically sharp interface enabled ultrahigh-speed non-volatile memory devices. *Nat. Nanotechnol.* **16**, 882–887 (2021).
2. Liu, L. et al. Ultrafast non-volatile flash memory based on van der Waals heterostructures. *Nat. Nanotechnol.* **16**, 874–881 (2021).
3. Huang X et al. An ultrafast bipolar flash memory for self-activated in-memory computing. *Nat. Nanotechnol.* **18**, 486-492 (2023).

Revision:

In this revision, we've included discussion of the mentioned paper in the introduction part, and also include it into the literature comparison in **Table S1** and **Extended data Figure 1**.

At page 2 Line 24-26 in the revised manuscript.

“To this end, a recent example that adopts bipolar WSe₂ as the channel displays the potential in achieving well-balance memory performances by combine the Lucky-electron injection mechanism, with an oxide charge trapping layer structure for endurance enhancement.³²”

At page 8 Line 4-7 in the revised manuscript.

“In addition, replacing graphene with MoS₂, a symmetric tunneling structure of MoS₂/hBN/ MoS₂ with respectively edge and top contact to channel and float gate displays highly asymmetric P/E speed, showing an ultrafast program within 10 ns but slow erase >100 ms. This again confirms the edge contact directly contributes to the charge injection (**Supplementary Information S6-5**)”

Table S1. State of art performance of 2D flash memory compared to silicon flash.

	Functional layer	Voltage	Speed	Retention	Endurance (SLC)	Endurance (MLC)
2D flash memory	1T-2H-MoS ₂ /hBN/Gr (This work)	±15 V	100 ns	>10 ⁴ s (>10 year by acceleration, 85 °C)	>10 ⁶	> 8x10 ⁴
	WSe ₂ /hBN/HfO ₂ /Al ₂ O ₃ ¹⁴	±30 V	50 ns	>10 ³ s (>10 year by acceleration, 150 °C)	>10 ⁶	N. A.
	MoS ₂ /hBN/Gr ¹⁵	±30 V	20 ns	>10 ⁵ s (>10 year by acceleration, 85 °C)	>1390	N. A.
	InSe/hBN/Gr ³	-20.8 V +20.2 V	21 ns	>10 ³ s (>10 year by extrapolation)	>2000	N. A.
	MoS ₂ /hBN/Gr ¹⁶	±15 V	100 μs	>1400 s (>10 year with extrapolation)	>100	N. A.
	MoS ₂ /hBN/Au ¹⁷	±5 V	100 ms	>10 ⁵ s (>10 year by extrapolation)	>10 ⁵	N. A.
	MoS ₂ /HfO ₂ /Gr ¹⁸	±15 V	100 ms	>2000 s (>10 year by extrapolation)	>120	N. A.
	BP/Al ₂ O ₃ /Al ₂ O ₃ ¹⁹	±20 V	100 ms	>10 ³ s (2x10 ⁷ by extrapolation)	>100	N. A.
	MoS ₂ /hBN/BP ²⁰	±20 V	300 ms	>10 ³ s	>50	N. A.

Extended Data Figure 1. Comparison of the speed-endurance characteristic of present edge contacted flash memory as SLC to previous 2D flash memory and silicon flash memory.

Q2:

We have noticed that the reported robust endurance work (Nat. Nanotechnol. 18, 1–7; 2023) has not been cited in the performance comparison chart (extended data figure 1).

Reply:

We thank the reviewer’s comments. We’ve included the mentioned paper in the chart. Please refer to reply to Q1.

Revision:

In this revision, we’ve included the discussion to above mentioned literature in **Extended Data Figure 1** and **Table S1**.

Q3.1:

Another major drawback of this work is that the justification for the mechanism is confusing: The authors have claimed that the 1T-MoS₂ edge-contact is much superior to the conventional top-contact device. However, all the contact region of the edge-contact device are not well defined and all these so-called edge-contact devices have actually both top-contact and edge-contact with the channel materials. The authors should provide clean experimental results to support their point of view.

Reply:

We thank the reviewer’s comment. We’d like to state that although both contacts coexist in our devices, we have demonstrated that top contact exhibit apparently low tunneling current (1T-

MoS₂/hBN/Graphene) if compared to the tunneling through semiconductor channel (2H-MoS₂/hBN/Graphene) (**Figure S6**). To further validate the role of edge contact, we provide in this revision direct evidence on the $>10^2$ enhanced tunneling current in edge contacted heterostructure compared to top contact configuration at the field strength of 8MV/cm. (**Figure S6-4**). The role of edge contact in endurance enhancement is also supported by the added time-to-breakdown analysis, which is suggested by reviewer 3 in Q4. These results combined with our previous experiment evidence clearly validated the critical role of edge contact in improving the overall memory performance, and achieving a well-balanced speed and endurance lifetime.

Figure S6-4. Comparison of the transient tunneling current of memory cells with edge and top contact. **a**, The schematic diagram of the transient tunneling current test. **b**, The demonstration of the transient voltage applied and current monitored. The vertical voltage is applied within 1 ms and the monitor of the corresponding tunneling is also finished, and the comparison of tunneling current density are normalized according to the device area (**c**).

In addition, we'd like to note that the adopted contact configuration is indeed deliberately designed to avoid undesired lithium intercalation in the heterostructure. As illustrated in **Figure R1**, for gate modulation to MoS₂ channel by the float gate, intercalation in the interlayer between MoS₂ and hBN tends to screen electric field and deteriorate device performance. This is apparent when we compare the transistor performance that is processed with exposed or sealed edge of MoS₂/hBN. The adopted configuration in present work allows us to dramatically optimize subthreshold swing (SS), which reflect that efficient coupling between float gate and the 1T-2H Schottky contact in the MoS₂ channel is formed. To realize ultrafast memory speed, such contact configuration is necessary at present stage, however, further optimization will be possible if intercalation in heterostructure is avoided. The results will be discussed in our on-going work.

Figure R1 (for review only). Design and improvement of 2D edge contact for floating-gate transistor. (a, b) Schematic diagrams of atomic structure (left) and band structure (right) of lateral type (a) and top to down type (b) lithium intercalation method. **c.** Gate-dependent current curves of the transistor constructed by lateral type (left) and top to down type (right) lithium intercalation method. **d.** Subthreshold swing statistics distribution of 6 transistors from both lateral type samples and top to down type samples.

Revision:

In this revision, we provide additional analysis based on direct measurement of tunneling current (**Figure S6-4**). At Page 7 Line 30 to Page 8 Line 4 in the revised manuscript.

“However, by comparing directly the tunneling current across edge contacted 2H-MoS₂/hBN/graphene heterostructure, we found ~2 orders enhancement to conventional top contacted one at the electric field of 8 MV/cm (**Supplementary Information S6-4**). This clearly suggest that the edge contact interface play vital role in enhancing the charge injection efficiency.”

And discussion about our device configuration has been added at page 4 Line 17-19 in the revised manuscript:

“In our design, the phase transformation in MoS₂ is ensured to reach the bottom layer from the top surface, which avoids intercalation at the hetero-interface, e.g., the one between MoS₂ and hBN.”

Q3.2:

The authors said the ideal Schottky contact is important, but no characterisation directly demonstrates that the 1T-MoS₂ are ideal Schottky contacts without any degradation. What is the exact height of the Schottky barrier here? How does this affect the performance?

Reply:

We thank the reviewer's comments. We have extracted the Schottky barrier from both edge and top contacted devices, using temperature dependent measurements (**Figure S13-1**), and analyzed the gate modulation to Schottky barrier in both devices (**Figure S13-2**). The results indicated that edge contact provided a more tunable Schottky barrier to MoS₂ channel, which is critical for the hot carrier injection through Schottky contact under field effect modulation. The results are also consistent with our CV analysis on the defect density in edge and top contact, and consistently supported our discussion to the role of contact.

In our discussion, we attribute the speed enhancement by edge contact to the less defect density at the contact interface. This would make the Schottky barrier in edge contact highly tunable under gate modulation. As direct support, we've extracted the Schottky barrier for different contact modes at varying gate bias, using temperature dependent measurements, which are shown in the following **Figure S13-1**. We note that due to the n-type behavior of MoS₂, the barrier corresponds to electron barrier Φ_{SB}^e , from what the hole barrier can be calculated as $\Phi_{SB}^h = Eg - \Phi_{SB}^e$.

The highly tunable Schottky barrier in edge contact is reflected by Schottky barrier (Φ_{SB}^e) under gate modulation. As shown in the following **Figure S13-2**, edge contact to 2H-MoS₂ exhibits a highly tunable (Φ_{SB}^e) with $d\Phi_{SB}^e/dV_{FG} = -0.21$, while top contact is in a strongly pinned state with $d\Phi_{SB}^e/dV_{FG} = -0.02$. Such highly tunable Schottky barrier of edge contact means strong energy band bending in MoS₂ part under gate modulation, which is essential to enable hot hole injection into channel via field effect emission process. If the Schottky barrier at contact is pinned by high density interfacial states, e.g., in the case of top contact, most of the carriers are injected via thermionic process, which would not enhance the tunneling efficiency to float gate.

Figure S13-1. Comparison of temperature-dependent transfer curves with floating gate in device MBG1# and PMBG8# and the Arrhenius fitting curves for the extraction of Schottky

barrier height. (a, b), The transfer curves of the top/edge contacted flash memory devices while the temperature changes from 80 K to 330 K with a step of 25 K. (c, d), Arrhenius fitting curves at different floating gate voltages in device MBG1# and PMBG8#, respectively.

Figure S13-2. The extracted effective barrier height (Φ_{SB}^e) as a function of applied floating gate voltages for top contact and edge devices, and the FLP factor was extracted.

Revision:

In this revision, we provide additional analysis based on temperature-dependent transfer curves of top contact and edge contact device (**Figure S13-1**) and extracted the Schottky barrier for them (**Figure S13-2**).

We've added the following discussion about the Schottky barrier at Page 9 Line 3-6 in the revised manuscript.

“In present float gate memory cells, the reduced trap density in edge contact renders highly tunable Schottky barrier under gate modulation (**Supplementary information S13**), which is essential for hot carrier injection via Schottky emission at the metal-semiconductor interface.”

Q3.3:

The 2H-MoS2 channels are known as n-type semiconductors and the metallic contacts are also full of electrons. But the energy band diagrams in Figure 3a&b were drawn from the point of hole conducting. This is contradictory.

Reply:

We thank the reviewer's comments. We understand the confusion from the reviewer on the type tunneling carrier, which is still a topic in debate for 2D flash memory. However, we'd like to clarify that our illustration does not contradict to prevail understanding. If compared to two recent papers, [1, 2] there is apparent difference in the illustrated memory operation. Typically, for either program or erase, opposite carriers shall be involved. In previous reports, Wu and Liu et. al. explained erase operation ($1 \rightarrow 0$) of the memory using electron tunneling. [1, 2] However, in our case, we choose to

illustrate program operation (0→1) considering that our edge contact has more dramatic impact to program speed if compared to erase. For program, either hole tunneling from MoS₂ to graphene or reverse electron tunneling from graphene to MoS₂ is responsible. According to our previous discussion in **Figure S6-2** based on gate modulated tunneling current, we are persuaded that hole tunneling is responsible in this program operation.

Although MoS₂ is a typical n-type semiconductor, given the significant electric field of hBN, the generation of hole inversion is possible. For example, considering 10 V potential drop across 10nm hBN, the induced positive charge would be at the level of $2 \times 10^{13} \text{ cm}^{-2}$ (*charge density* $\sigma = \epsilon_0 \epsilon_{hbn} E$, *density of holes* $= \sigma/e$), which easily exceed the inherent donor concentration (up to 10^{11} cm^{-2}) in MoS₂. Recently, there are other reports that discussed hole tunneling is responsible for the memory operation in many 2D flash memory cells, including MoS₂/BN/Gr, MoTe₂/BN/Gr, WSe₂/BN/Gr heterostructure based device. [3, 4] They've reached this conclusion based on the analysis of gate modulation to tunneling current (see following **Table R1**). We adopted a similar strategy in our investigation and reached consistent conclusion (**Figure S6-2**). Thus, we are confident that our illustration in **Figure 3c** and **d** reasonably reflects the memory operation.

Table R1 (for review only). The comparison of carrier polarity in FN Tunneling at hBN/Gr based heterostructure

Ref.	Channel material	Carrier under negative E	Carrier under positive E	Method
1	InSe	Electron	Electron	Band alignment
2	MoS ₂	Hole	Electron	Band alignment
3	Pd, Au, Cr and Ti	Hole	Hole	Gate-dependence Tunnel current (TC) test & simulation
4	MoTe ₂ , MoS ₂ and WSe ₂	Hole	Hole	Gate-dependence TC test & channel-dependence TC test
5	MoTe ₂	Hole	Hole	Gate-dependence TC test
Our work	1T-2H MoS ₂	Hole	Hole	Gate-dependence TC test & Band alignment (Figure S6)

References:

1. Wu, L. et al. Atomically sharp interface enabled ultrahigh-speed non-volatile memory devices. *Nat. Nanotechnol.* **16**, 882–887 (2021).
2. Liu, L. et al. Ultrafast non-volatile flash memory based on van der Waals heterostructures. *Nat. Nanotechnol.* **16**, 874–881 (2021).
3. Hattori, Y et al. Determination of Carrier Polarity in Fowler–Nordheim Tunneling and Evidence of Fermi Level Pinning at the Hexagonal Boron Nitride/ Metal Interface. *ACS Appl. Mater. Interfaces*, **10**, 11732-11738 (2018).
4. Sasaki T et al. Material and Device Structure Designs for 2D Memory Devices Based on the Floating Gate Voltage Trajectory *ACS Nano*, **15**, 6658-6668 (2021).

5. Sasaki T et al. Understanding the Memory Window Overestimation of 2D Materials Based Floating Gate Type Memory Devices by Measuring Floating Gate Voltage. *Small* **16**(47): e2004907 (2020).

Revision:

In this revision, additional explanation related to the illustration to memory operation is provided to avoid confusion. The related discussion is included at page 7 Line 18-25 in the revised manuscript.

“Previously, hole tunneling had been discussed responsible for the operation of 2D float gate transistors that use MoS₂ as the channel and graphene as float gate.³⁶ In our case, the dramatic enhancement of P/E speed via contact engineering to MoS₂ suggest that electron/hole tunneling from MoS₂ to graphene is most likely responsible for the memory operation. Here, hole tunneling is used for illustration considering the apparent improvement to program speed if compared to erase. Although MoS₂ displays n-type conduction with electrons as the majority carriers, the high field modulation under positive gate bias during program is sufficient to induce significant amount of holes at the level of 10¹³ cm⁻² if considering a field strength of 10 MV/cm across hBN.”

The experiment design in this work also has obvious flaws:

Q4:

First, even though the paired devices were fabricated on the same vdW heterostructure, the obtained 1T (edge contact) region is just defined in a little square and there are still conventional top electrode contacts with the 2H-MoS₂ channel outside the 1T square (Figure 2b). Therefore, the experimental comparison of the memory cells with 1T edge or top-contacts is inconsistent. Because the 1T edge contacted devices actually possessed both 1T edge and top contacts.

Reply:

We thank the reviewer's comments. Due to its correlation to previous raised concerns in Q2, detailed reply to the comment is referred to Q2.

In brief, we'd like to state that Cr/Au electrodes were only used to connect the phase transformed region of 1T-LixMoS₂. Although Cr/Au contact 2H-MoS₂ in part area, our evidence shows that its tunneling current density is significantly lower if compared to the tunneling through 2H-MoS₂ channel. In addition, we mention that the adopted contact configuration is required in our work to improve the float gate coupling to the Schottky barrier at 1T-2H contact. To avoid the effect to Cr/Au contact region in edge contacted devices, we intentionally control its area relative to edge contact <10%.

Revision:

In this revision, we've provided additional discussion to experiment design in the experiment and method part.

At page 4 Line 17-19 in the revised manuscript:

“In our design, the phase transformation in MoS₂ is ensured to reach the bottom layer from the top surface, which avoids intercalation at the hetero-interface, e.g., the one between MoS₂ and hBN.”

At page 11 Line 15-19 in the revised manuscript:

“We attribute the superior endurance lifetime to its better time to breakdown performance related to hBN layer in edge contact configuration (**Supplementary Information S18**), which suppress interfacial roughness arisen from lattice distortion in MoS₂ by direct metal evaporation. The present strategy transforms 2H-MoS₂ under metal contact into its metallic 1T phase, it's ultraflat as an interfacial layer.”

Q5:

Further, according to the fabrication detailed descriptions the top contacts were made after the 1T contacts. So the top contacts underwent more additional manufacturing steps than 1T contacts. If there is no particular process to remove PMMA residues to ensure a clean interface, there are no surprises that the top contacts with poor interfaces have a worse performance than the edge contacts. In other words, the comparison between the different contacts is self-referential and unconvincing to the scientific community due to the lack of a rigorous experimental design.

Reply:

We thank the reviewer's comments. We'd like to state that our conclusion is not biased or self-referential, but is consistently supported by our experimental results from multiple aspects, including paired memory cells, statistical analysis, and direct tunneling current measurements. We'd like to clarify that the reason why we chose paired memory cell in discussion, is to avoid uncertainties caused by the material quality, interface quality, and thickness variation related to hBN or MoS₂. In previous discussion, the quality of vdW interface had been evaluated highly for achieving a fast memory operation.

As to the mentioned contamination effect, in experiment, we did not observe apparent performance degradation for paired top contacted memory cell if compared to those experiences only one lithography step. Furthermore, our statistical comparison in **Figure 2e and f** for top and edge contact devices is based on >40 devices made from identical lithography processes, this could undoubtedly support our discussion, that is edge contact could indeed improve the key performance of 2D flash memory.

Q6:

For the 10 years retention test, we advice that the author should measure the data retention ability at least 3 different temperatures to fit the data retention time at room temperature.

Reply:

We thank the reviewer's comments. In this revision, we have added the additional retention evaluation based on measurements at 5 different temperatures (55 °C, 85 °C, 105 °C, 125 °C, 145 °C), as shown in following **Figure S15**. In the temperature accelerated evaluation, the retention

lifetime is evaluated using Arrhenius plot method (**Figure S15c**), from what we obtain the estimation that the data can be stored for 10 years at a temperature of 78.6 °C. Similar method had been used in ref. [1], instead we use the extrapolation of both on and off state to extract the retention lifetime. The standard is stricter if compared previous extrapolation of on state to a constant reference state.

Figure S15. Temperature-dependent retention performance and the extraction of activation energy by Arrhenius plot method. a, Retention performance at each field temperature (55, 85, 105, 125, 145 °C) in the edge contacted flash memory device PMBG17#. **b**, Retention-failure evaluation of the device from the V_{th} at different temperatures. **c**, Arrhenius fitting and extrapolation for the retention lifetime. **d**, The calculated acceleration ratio under different temperature.

Reference:

1. Liu, L. et al. Ultrafast non-volatile flash memory based on van der Waals heterostructures. *Nat. Nanotechnol.* **16**, 874–881 (2021).

Revision:

In this revision, we have provided the temperature dependent retention measurements (55, 85, 105, 125, 145 °C) in **Figure S15** for the evaluation of retention lifetime.

Reviewer: 2

By adopting metallic 1T-LixMoS₂ as edge contact, the authors we show that ultrafast (10-100 ns) and super robust (endurance>10⁶ cycles, retention>10 years) memory operation can be simultaneously guaranteed in two-dimensional van der Waals heterostructure flash memory with 2H-MoS₂ as semiconductor channel. The superior performance stems from the highly gate tunable Schottky barrier at the edge contact, which facilitates hot carrier injection to semiconductor channel and subsequent tunneling if compared to conventional top contact that has rich defects under metal.

For publication in Nature Electronics, the authors must show that this approach is novel and unique. I understand that other groups have tried edge contacts already. The authors need to do a more comprehensive review of other work on 2D devices to show this work is not incremental. The main innovation appears to be the edge contact and more details need to be given.

Reply:

We thank the reviewer's comment. We understand that edge contact had been previously proposed for 2D transistors, but their potential benefits and role in 2D flash memory had not been discussed. Based on the novel contact engineering strategy for 2D flash memory, our work has achieved the enhancement in speed and endurance simultaneously, which is a highly challenging task due to the dilemma of "speed-endurance trade-off". At 2021, two reports achieved ultrafast speed at 20 ns level based on the vdW heterostructured flash memory. [1, 2] However, a competing endurance lifetime at that moment is not demonstrated. By using a new contact engineering strategy based on in-situ phase change, our work achieved simultaneously ultrafast memory operation within 10-100 ns while having a competing endurance lifetime that meet the standard of commercial silicon flash.

We note that our strategy is different with the mentioned publication in Nat. Nanotechnol. 18, 1–7; 2023, [3] which is published during the review process of our manuscript. The report exploits Lucky-electron tunneling and a charge trapping stack of hBN, HfO₂ and Al₂O₃ to improve speed and endurance respectively. Differently, our work exploited the contact engineering strategy to achieve this, and our experiments consistently show that contact interface plays critical role in improving the overall performance of 2D flash, and replacing conventional top contact using phase engineered edge contact provides at the moment the most competing performance, which has superior P/E speed while fulfilling the requirement for endurance in commercial flash memory. Our strategy based on contact engineering had not been reported at the moment, and we believe it is novel and may open a new horizon to further engineering 2D flash memory. For clarity, we've improved our discussion at introduction part to highlight our innovation.

Reference:

1. Wu, L. et al. Atomically sharp interface enabled ultrahigh-speed non-volatile memory devices. *Nat. Nanotechnol.* **16**, 882–887 (2021).
2. Liu, L. et al. Ultrafast non-volatile flash memory based on van der Waals heterostructures. *Nat. Nanotechnol.* **16**, 874–881 (2021).

3. Huang X et al. An ultrafast bipolar flash memory for self-activated in-memory computing. *Nat. Nanotechnol.* **18**, 486-492 (2023).

Revision:

In this revision, we included the following discussion to highlight the innovation in the proposed edge contact strategy for improving the performance of 2D flash memory.

At Page 2 Line 26-30 in the revised manuscript:

“On the other hand, though an ideal Schottky contact could theoretically enhance charge injection efficiency in a flash memory, the vital role of contact interface in leveraging memory performance had been overlooked in the past, especially considering that the ultrathin two-dimensional crystal lattice is extremely sensitive to direct metal deposition in the conventional top contact configuration.”

At Page 3 Line 6-8 in the revised manuscript:

“This markedly improves the charge injection efficiency to float gate and guarantees ultrafast and super-robust memory operation at the same time, which was been rarely reported in 2D flash memory.”

Q1:

Abstract

“flash memory ... typical operation speed is limited ~microseconds to milliseconds for program and erase”

- Need to verify as I understand it might be ns now

Reply:

We thank the reviewer’s comment. After a comprehensive investigation of the literature and industrial technical manuals. The performance of prevailing Si flash is confirmed at the level of microseconds to milliseconds. For example, in the discussion of ref.1 [*Nat. Nanotechnol.* **16**, 882–887 (2021).]: ‘..., typical write times of hundreds of micro- or milliseconds remain a few orders of magnitude longer than that of their volatile counterparts’.

For the performance of commercial flash memory in the form of NAND or NOR, we’ve performed survey from literature, [1] open manuals and also latest study. For example, in the case of commercial NAND flash, the disclosed program speed is 300-700 μs (Micron Technical Note TN-29-14, see webpage or the uploaded file for review: **FR1**), [2, 3] For NOR flash memory, it has faster program speed by adopting hot carrier injection from lateral electric field acceleration under source-to-drain bias. Their program speed could reach $\sim 10 \mu\text{s}$, while the erase still relies on FN tunneling and displays typical erase time $\sim 20 \text{ ms}$ (SuperFlash, by Microchip, see webpage or uploaded file for review: **FR2**). [4] Therefore, the description of typical operation speed flash memory \sim microseconds to milliseconds is reasonable.

References:

1. Richter, D. Flash Memories: economic principles of performance, cost and reliability optimization, Springer, 2014.
2. Increasing NAND Flash performance, Micron, webpage or uploaded file for review: **FR1**.
3. Hidaka, H. Embedded Flash Memory for Embedded Systems: Technology, Design for Sub-systems, and Innovations, Springer, 2018.
4. SuperFlash Memory Products, Microchip, webpage or uploaded file for review: **FR2**.

Q2:

p.2 “the performance of 2D flash memory in previous literatures falls behind of the expectation in spite of various channel and float gate used. Until recently, 20-160 ns superior operation speed was achieved in InSe and MoS₂ flash memories based on the clean interface in vdW heterostructures or hot carrier injection directly through the ultrathin 2D material. However, a competing endurance lifetime to the prevailing Si flash technology (>10⁵ cycles) had not been demonstrated for a simultaneously ultrafast and robust flash memory.”

- what is the new innovation here?

Reply:

We thank the reviewer's comments. In this work, a novel strategy based on contact engineering is proposed to optimize the key performances of 2D flash memory, covering the overall requirements to speed, retention and endurance. Our study shows clearly that if compared to conventional top contact, having a phase engineered edge contact renders a highly tunable Schottky barrier at the contact interface, which allows injection of hot carriers through field emission. Due to the unique edge contact, the operating mechanism of our ultrafast 2D flash memory is attributed to hot carrier generation in Schottky contacted field-effect transistor, which provides highly enhanced charge injection efficiency if compared to conventional FN tunneling. Besides, different from previous work [1, 2] that focus on achieving ultrafast speed, our work guarantees a robust P/E cycle >10⁶ cycles while having simultaneously ultrafast speed. This is because given the superior charge injection efficiency, we could achieve ultrafast memory operation at lower voltages, which extended memory lifetime. We believe our strategy points out a new direction to further engineer 2D flash memory, and could greatly promote the understanding in this field.

References:

1. Liu, L. et al. Ultrafast non-volatile flash memory based on van der Waals heterostructures. *Nat. Nanotechnol.* **16**, 874–881 (2021).
2. Wu, L. et al. Atomically sharp interface enabled ultrahigh-speed non-volatile memory devices. *Nat. Nanotechnol.* **16**, 882–887 (2021).

Revision:

In this revision, we included the following discussion to highlight the innovation in the proposed edge contact strategy for improving the performance of 2D flash memory.

At Page 2 Line 26-30 in the revised manuscript:

“On the other hand, though an ideal Schottky contact could theoretically enhance charge injection efficiency in a flash memory, the vital role of contact interface in leveraging memory performance had been overlooked in the past, especially considering that the ultrathin two-dimensional crystal lattice is extremely sensitive to direct metal deposition in the conventional top contact configuration.”

At Page 3 Line 6-8 in the revised manuscript:

“This markedly improves the charge injection efficiency to float gate and guarantees ultrafast and super-robust memory operation at the same time, which was been rarely reported in 2D flash memory.”

Q3:

Fig.1 – can a HRTEM image of an edge contact be shown? What proof is there of edge contact?

Reply:

We thank the reviewer’s comments about the atomic structure of 1T/2H interface. According to the reviewer’s suggestion, we’ve replaced previous TEM image using the following HAADF-STEM image. In following **Figure 1b**, the atomic structure of the heterostructure is observed along the MoS₂ zig-zag direction. The phase transition interface of MoS₂ is analyzed with the corresponding atomic model shown below. The overlaid blue and red dashed lines highlight the lattices of hexagonal MoS₂, octahedral MoS₂, respectively.

Figure 1b Atomic-scale observation of the edge contact in MoS₂, the high-angle annular dark field

scanning transmission electron microscopy (HAADF-STEM) image validates the phase transformation of 2H MoS₂ into a distorted 1T phase.

Revision:

In response to reviewer's concern, we've replaced previous TEM image in **Figure 1b** using a new HAADF-STEM image.

The related discussion is included at page 4 Line 12-17 in the revised manuscript:

“The vdW interfaces in the fabricated heterostructure are confirmed via the high-angle annular dark field scanning transmission electron microscopy (HAADF-STEM) image (**Figure 1b**). A transition from 2H-MoS₂ in channel to a distorted 1T-MoS₂ structure could be identified near the contact pad, forming the lateral edge contact. Due to the interlayer diffusion of lithium, the 1T-2H interface extrudes slightly from the lithography patterned area in to the channel, and has slight structure distortion in plane due to intercalation induced strain.”

Q4:

Fig.2 – the difference between 1T edge and Cr top contacts could be more clearly shown, as in HRTEM images.

Reply:

We thank the reviewer's comments. According to the reviewer's suggestion, the 1T edge contact is already confirmed by HAADF-STEM image in **Figure 1b**, and the details of discussion are provided in previous reply to Q3. To clearly reveal the difference of edge and top contact mode, we compare HRTEM images at different region in device. As shown in the following **Figure R2a** and **b**, under the deposited Cr/Au contact, both configuration exhibit lattice disorder due to the penetration of foreign atoms into the lattice of 2D MoS₂. However, in the case of edge contact by Li intercalation, the metallic 1T-MoS₂ extrude slightly into the channel, avoiding direct damage to the 1T/2H-MoS₂ contact interface by deposition processes. Our results on CV analysis (**Figure 3c**) clearly indicated that such method suppresses interface trap density in devices, which is consistent with the suppressed structure disorder at edge contact interface.

Figure R2 (for review only). TEM characterization for MoS₂/hBN/FLG vdW heterostructure. a, The structure diagram (top) and cross-section TEM image (bottom) of the MoS₂/hBN/FLG heterostructure under the deposited Cr/Au electrode, adapted from **Figure S12-3b**, The structure diagram (top) and HAADF-STEM image of the MoS₂/hBN/FLG heterostructure under (left of bottom) and outside (right of bottom) the deposited Cr/Au electrode.

Q5:

Fig.3 – a schematic is shown for the edge contact, but no HRTEM image is shown. Show it to prove an edge contact is truly formed.

Reply:

We thank the reviewer's comments. According to the reviewer's suggestion, we've replaced previous TEM image using a HAADF-STEM image in **Figure 1b**. The details of discussion are provided in previous reply to Q3.

Q6:

p.11 “~1 order better than the lower limit of commercial silicon flash memory (~10⁵ cycles for SLC),” - this not a fair comparison (with lower limit?)

Reply:

We thank the reviewer's comments. We have revised our description from 'the lower limit' to 'typical' in revised manuscript. After a comprehensive investigation of the literature and industrial technical manuals. We confirmed that the industry generally considers 100,000 P/E cycles as the typical (TYP) standard for endurance lifetime, e.g., Macronix SLC NAND Flash memories in Ref. 1. In our case, we achieved ultrafast P/E speed ~10-100 ns while having excellent endurance

lifetime $>10^6$ cycles. By guarantee such endurance lifetime, the speed for P/E can be dramatically increased. For a proper and justified comparison to commercial standard, we have surveyed lot of commercial flash memory for single-level-cell (SLC), and summarized their performance in the following **Table S2**.

Table S2. State of art performance of 2D flash memory compared to commercial flash.

Memory type		Program time	Erase Time	Endurance	Ref.
NAND	MPN: MT29F2G08AABWP ²	300 μ s	2 ms	10^5	24
	MPN: MT29F16G08ABACA ³	350 μ s	1.5 ms	6×10^4	25
	MPN: MT29F2G08A ⁴	220 μ s	500 μ s	10^5	26
NOR	MPN: TE28F128J3 ⁴	128 μ s	1 s	10^5	26
	MPN: MT25QU256ABA ⁵	120 μ s	150 ms	10^5	27
	MPN: N25Q256A ⁶	500 μ s	250 ms	10^5	28
Serial Flash Embedded Memory	MPN: M25P128 ⁷	500 μ s	500 μ s	10^5	29
2D Flash	1T-2H-MoS ₂ /hBN/Gr	10 ns	100 ns	3×10^6	This work

**MPN: Manufacturer Product Number*

References:

1. AN0339V1-Endurance and Retention of NAND Flash, Macronix, webpage or uploaded file for review: **FR3**.
2. 2, 4, 8Gb: x8/x16 Multiplexed NAND Flash Memory Features, Micron, webpage or uploaded file for review: **FR4**.
3. 16Gb, 32Gb, 64Gb Asynchronous/Synchronous NAND Features, Micron, webpage or uploaded file for review: **FR5**.
4. TN-29-19: NAND Flash 101: An Introduction to NAND Flash and How to Design It In to Your Next Product Micron, webpage or uploaded file for review: **FR6**.
5. 256Mb, 1.8V Multiple I/O Serial Flash Memory Features, Micron, webpage or uploaded file for review: **FR7**.
6. 3V, 256Mb Multiple IO Serial Flash Memory Features, Micron, webpage or uploaded file for review: **FR8**.
7. M25P128 Serial Flash Embedded Memory Features, Micron, webpage or uploaded file for review: **FR9**.

Revision:

To avoid confusion, we've revised the following discussion in revision at page 11 line 10-11.

"It is worth noting that the attained endurance lifetime achieved in edge contacted memory cells is ~1 order better than the typical standard of commercial silicon flash memory (~ 10^5 cycles for SLC),⁴¹⁻⁴⁴"

Q7:

p.11 "Such performance is ~1 order better than the commercial standard (~ 3×10^3 cycles for MLC) while having the ultrafast P/E speed. In the future, by reducing the channel length and adopting vertical 3D packing based on the feasible vdW stacking of 2D materials, the bit density of such vdW stacked flash memory can be markedly improved to meet the high volume density of existing silicon flash."

- this has not been proven yet (reducing the channel length and adopting vertical 3D packing)

Reply:

We thank the reviewer's comments. Since demonstrating a short channel or 3D stacking memory is beyond the scope of present work, we've revised the discussion by focusing on what need be further evaluated before realizing a high-density integration, including its application in a NAND or NOR architecture, and the potential challenges related to lithium intercalation.

Revision:

In response to reviewer's concern, we've revised the following discussion in revision at page 12 Line 2-6 in the revised manuscript.

"In the future, further study on the integration of such memory cells in NAND or NOR architectures could promote its application in high density mass-storage memory.⁴⁷ Meanwhile, for high density integration of a dramatically scaled memory cells with reduced channel and contact size, precise control to the position and quality of phase engineered contact would be critical"

Q8:

Please comment on the reliability of the devices, how many were made and how many worked.

Reply:

We thank the reviewer's comments. By analyzing 26 cells with 1T edge and 23 cells with Ct top contact, (**Figure 2e and f**), we found edge contact indeed dramatically improves the yield for ultrafast memory cells. For edge contact device, the value reached 90% for successful program within 10 ns, while it is 40% for successful erase within 100 ns. As to the endurance lifetime, edge

contacted memory cells also act better than conventional top contacts. We've confirmed multiple edge contacted devices showing long endurance lifetime $>10^5$ cycles (**Supplementary information S16-3**), while few top contacted memory cells passed 10^4 cycles.

Figure S16-3. Endurance performance of edge contacted flash memory cells (as SLC): a, PMBG4#; b, PMBG19#. The optical image and endurance characteristic are displayed for each cell. The endurance lifetime of edge contacted memory cells is $\sim 10^5$ - 10^6 cycles.

Reviewer: 3

In this manuscript, Yu et al. reported a novel edge contacted MoS₂ float gate transistor, which has impressive endurance lifetime $>10^6$ while operating at ultrafast speed ~ 10 -100 ns. They found that the adoption of 1T MoS₂ as edge contact apparently improves the device performance if compared to conventional face contacted metals (Cr/Au). Based on comprehensive investigations, such improvement was attributed to the Schottky injection at from edge contact to MoS₂, which then creates hot carriers in MoS₂ channel and enhances the tunneling probability through hBN layer. The device performance is impressive, considering the endurance lifetime is comparable to industry standard (10^5) for a robust mass storage memory, but at orders' faster P/E speed. Since previous investigations mostly focus on tunneling interfaces, the idea of engineering contact interface via edge contact is novel in boosting the performance of 2D flash memory, and may stimulate further research in this direction.

The manuscript is well organized, and provided discussion of statistical analysis of device variation and yield for ultrafast memory, which is of substantial importance to further leverage the study for novel flash memory via 2D materials, and potentially benefit innovation in its Si counterparts. Thus, I would like to recommend its publication if the following concerns can be well-addressed.

Q1:

1. In Figure 1c, a 10-ns pulse can switch the device to the high-current state, but the series data lacks intermediate programming process as in Figure 1d. The pulse-width-dependent program performance should be presented. One or more intermediate points in Figure 1c will be convincing.

Reply:

We thank the reviewer's comments. Since 10 ns pulse is the minimum pulse width that can be achieved using our pulse generator (Agilent 81110A), in the evaluation of P/E speed, we cover different pulse width from 10 ns to 100 ns. However, for most devices, the program speed is faster than erase, some of them can be fully switched within 10 ns. In this case, we provide results at lower operation voltages, in which the gradual switch can be revealed. The results are supplied in following **Figure R3**.

Figure R3 (for review only). Detailed test result for operation speed under different operation voltage (from ± 5 to ± 15 V). Figure 1c is adapted from Figure R3k and v, and the result of ON/OFF ratio under variable pulse amplitudes and durations in Figure S10e is also extracted from here.

- Q2:**
2. The enhancement of charge injection efficiency is critical to boost the P/E speed at low operation

voltage. Therefore, it is important to clarify the tunneling pathway in the device. However, as the float gate, the graphene in the presented devices overlaps with metal contact, which makes direct tunneling pathway from metal to graphene possible, in addition to the presented pathway in Figure 3. If the claimed edge injection theory via edge contact is dominant, underlap of float gate to MoS₂ channel will deteriorate the injection efficiency even if 1T MoS₂ is used as the contact. The authors should compare 1T contacted devices with and without overlap between float gate and contact.

Reply:

We thank the reviewer's kind suggestions on the experiment. As shown in following **Figure S14**, we provided the comparison using paired memory cells which are made on the same vdW structure but with different overlap condition for fair comparison according to the suggestion. For the cell without overlap region, the extra region cannot be modulated by the gate electric field. This result in lower ON-state current in following **Figure S14c** and **d**. Their P/E performance is displayed in following **Figure S14e** and **f**. If taking an on/off ratio of 10^2 as the criteria for successful P/E operation, the cell with overlap region has faster P/E speed than the cell without overlap region at the same operation voltage. This is because only when graphene float gate overlaps with edge contact, the Schottky barrier at contact interface can be tuned by external gate bias, then contributing to the enhanced charge injection efficiency by hot carrier injection.

Figure S14. Comparison of the P/E performance of memory cells with and without overlap between float gate and 1T contact. (a, b), Optical microscope image (a) and schematic illustration (b) of the paired memory cells on the same vdW heterostructure. (c, d), Transfer curves under floating gate (c) and control gate (d) modulation. (e, f), Map of the attained ON/OFF ratio of memory under different voltage pulse conditions when changing both the amplitude and pulse width: 1T edge contact with overlap region (e) and without overlap region (f)

Revision:

According to the reviewer's suggestion, we've added the results in **Supplementary information S14**. The related discussion is included at Page 9 Line 6-10 in the revised manuscript.

“Consistently, when the edge contact is placed away from float gate coupling (**Supplementary information S14**), the absence of Schottky barrier modulation at the metal contact interface during memory operation deteriorates apparently the charge injection efficiency by edge contact. This validates that gate coupling to edge contact interface is crucial for achieving a superior charge injection efficiency.”

Q3:

3. The authors presented comparison of P/E speed of devices made of same vdW heterostructure and different contact configurations, including 1T edge contact and Cr/MoS₂ contact, and found the improvement of device performance by the adoption of edge contact. Following the above concern, if hot carriers are injected via edge contact, the authors should be able to observe direct enhancement of tunneling current in device. This is straightforward evidence that would support discussion in Figure 3. Hence, I suggest the authors to perform such measurement and discuss the results.

Reply:

We thank the reviewer's suggestions. Accordingly, we've performed direct measurement to tunneling current through edge and top contacted MoS₂/hBN/graphene heterostructure. To properly reflect the memory operation condition, we use fast IV measurement (by Agilent B1530, the waveform generator/fast measurement Unit) to probe the tunneling current within ms and avoid dielectric breakdown in hBN. Following **Figure S6-4** displays the time evolution of voltage and measured current for edge contact (EC) and top contact (TC). The current density through the heterostructure is estimated according to the device area, while the electric field in hBN is determined according to the thickness of hBN. The results for edge and top contacted heterostructure are compared in following **Figure S6-4c**, from what we do observe apparent enhancement to tunneling current by ~2 orders at the electric field ~8 MV/cm, which support our discussion of edge contact enhanced charge injection efficiency.

Figure S6-4. Comparison of the transient tunneling current of memory cells with edge and top contact. **a**, The schematic diagram of the transient tunneling current test. **b**, The demonstration of the transient voltage applied and current monitored. The vertical voltage is applied within 1 ms and the monitor of the corresponding tunneling is also finished, and the comparison of tunneling current density are normalized according to the device area (**c**).

Revision:

According to the reviewer's suggestion, we've added the results in **Supplementary information S6-4**. The related discussion is included from Page 7 Line 30 to Page 8 Line 4 in the revised manuscript.

"However, by comparing directly the tunneling current across edge contacted 2H-MoS₂/hBN/graphene heterostructure, we found ~ 2 orders enhancement to conventional top contacted one at the electric field of 8 MV/cm (**Supplementary Information S6-4**). This clearly suggest that the edge contact interface play vital role in enhancing the charge injection efficiency."

Q4:

4. Since 2D heterostructures have clean interfaces, what is the reason of device failure during P/E of 2D float gate memory? If it is determined by dielectric breakdown of hBN layer, I would like to see a time to breakdown analysis for hBN layer in the discussed different contact configurations, at comparable operation voltages. Will edge contact provide better interface with hBN layer and contribute to the endurance lifetime?

Reply:

We thank the reviewer's comments. The failure of flash memory as a charge trap memory usually occurs progressively in the form of trap creation and impact ionization in the tunneling layer under electric stress effect [1,2], and it completely breaks when the dielectric breakdown. According to the reviewer's suggestion, we have completed the experiment about time to breakdown (t_{BD}) analysis for hBN layer in the discussed different contact configurations.

As shown in following **Figure S18**, the evaluation is performed by incrementally increasing the electric field using 10 ns pulse, and then increasing the pulse duration from 10 ns to 100 ms. If the device passed time-to-breakdown test at 100 ms, the electric field is further increased to evaluate the limit. Our results clearly suggest that edge contacted heterostructure can tolerate higher electric

field under both the positive and negative bias.

Figure S18. Comparison of the electric field dependent time to breakdown of memory cells with edge contact and top contact device. **a**, Schematic of the test structure and bias condition for two type pulse condition. **b**, Pulse sequence in the test for positive and negative voltage stress. **c**, The time to breakdown versus E_{BN} data for the two type contact configurations. The hollow dot represent device could accept this pulse stress condition while the solid dot represent device is breakdown.

Reference:

1. Lee, J.-D. et al. Degradation of tunnel oxide by FN current stress and its effects on data retention characteristics of 90-Nm NAND flash memory cells. In *2003 IEEE International Reliability Physics Symposium (IRPS) 41st Annu* 497–501 (IEEE, 2003). doi:10.1109/relphy.2003.1197798.
2. Verweij, J. F. & Klootwijk, J. H. Dielectric breakdown I: A review of oxide breakdown. *Microelectron J.* **27**, 611–622 (1996).

Revision

According to the reviewer's suggestion, we've added the results in supplementary information **Figure S18**. The related discussion is included at Page 11 Line 15-19 in the revised manuscript.

"We attribute the superior endurance lifetime to its better time to breakdown performance

related to hBN layer in edge contact configuration (**Supplementary Information S18**), which suppress interfacial roughness arisen from lattice distortion in MoS₂ by direct metal evaporation. The present strategy transforms 2H-MoS₂ under metal contact into its metallic 1T phase, it's ultraflat as an interfacial layer.”

Q5:

5. High gate coupling ratio (GCR) close to 0.9 is exploited in this work, which is obtained by using an extended float gate structure that enlarges the capacitance between float gate and control gate. However, previous work usually reported GCR ~0.5, and such configuration sacrifices the area of device. How could such a high ratio be realized if high density integration is demanded in industry?

Reply:

We thank the reviewer's comments. About role of GCR, we'd like to clarify that GCR only influences the apparent operation voltage of the memory, and itself could not resolve the speed-endurance dilemma. It is the actual electric field in tunneling insulator that determines the speed and endurance limit of flash memory (rather than the operation voltage). The dilemma can only be overcome by improving charge injection efficiency, which is achieved in our case using hot carrier injection from phase engineered Schottky barrier transistor. The high GCR~0.9 of our devices come from the adoption of extended float gate terminal. We note that such structure exists ubiquitously in literatures, [1-3] and it help one to investigate the float gate coupling to semiconductor channel, thus the function of the memory.

About scalable integration, though the extended float gate configuration indeed allows one to realize a high GCR with relative ease, it does not necessarily lead to issue in making an integrated circuit. In commercial technology, high GCR~0.7 had been realized using asymmetric coupling between float gate to control gate and the semiconductor channel (see following **Figure R4** adapted from Ref. [4], in which the overlap between FG to semiconductor channel and CG are differently designed). Similar strategy had been also found in Micron flash memory (high coupling floating gate transistor, Patent No. 20040036108, see uploaded **FR10**). [5]

Redacted

Figure R4 (for review only). The asymmetric design used to optimize gate coupling ration in a flash memory cell. In this design, program gate (PG) to float gate (FG) was optimized to ~74%. Adapted from ref. [4]

References:

1. Paul, T. et al. A high-performance MoS₂ synaptic device with floating gate engineering for neuromorphic computing. *2D Mater.* **6**, 045008 (2019).
2. Wu, E. et al. Multi-level flash memory device based on stacked anisotropic ReS₂ –boron nitride–graphene heterostructures. *Nanoscale* **12**, 18800–18806 (2020).
3. Mukherjee, B. et al. Laser-assisted multilevel non-volatile memory device based on 2D van-der-Waals few-layer-ReS₂/h-BN/Graphene heterostructures. *Adv. Funct. Mater.* **30**, 2001688 (2020).
4. Houdt, J. V. et al. HIM0S-a high efficiency flash E²PROM cell for embedded memory applications. *IEEE Trans. Electron Devices* (1993). doi:10.1109/16.249473.
5. US patent, High coupling floating gate transistor, Paul Rudeck, 2003, See file uploaded for review: **FR10**.

Q6:

6. The adoption of lithium intercalation to obtain 1T MoS₂ can be also challenging in industry application. This is because lithium will likely diffuse in the interlayer of MoS₂ under the concentration gradient along the channel, how can the edge contact interface be controlled precisely, which is certainly important for the fabrication of highly integrated devices. The authors are

suggested to comment on the solution.

Reply:

We thank the reviewer's comment on this important issue. Achieving an ideally abrupt Schottky contact interface is highly demanded for generating hot carriers in a Schottky barrier transistor according to its mechanism. For previous Si technology, due to the processing limitation of silicide, the inter-diffusion of atoms, impurities degrade the interface quality and inhibited the progress toward an ideal Schottky contacted transistor.[1, 2] However, we note that by in-situ phase transformation from 2H MoS₂ to 1T, the interface can be atomically abrupt, given that phase transformation induced by lithium intercalation occurs very locally.[3] Though by lithium intercalation, the lithium diffusion is in-avoidable in the interlayer of MoS₂, the local characteristic of 1T phase transformation could still ensure an abrupt 1T-2H interface.

As shown in following **Figure R5a-c**, in experimentally, we do find that the formed phase transition interface exhibits some roughness beyond the patterning area defined by PMMA layer, which corresponds to the diffusion of lithium at interlayer space of MoS₂. To tackle this, we've developed a novel strain engineering strategy (**Figure R6**) to confine the lithium intercalation using step-edges of 2D materials, which closes the interlayer space of MoS₂ and increases the diffusion barrier of lithium. In our study, these steps displayed high blocking efficiency to the lithium intercalation if compared to the region without step-blocking. As indicated in following **Figure R5d-f**, for a step made by 10 nm hBN layer, the lithium intercalation induced phase transformation completely stops before the hBN step, while it extended apparently in bare MoS₂. In principle, the strategy could provide atomically abrupt phase change interface for an edge contacted memory cell. However, we are still developing the method, and it will be discussed in our later work.

Figure R5 (for review only). Strain engineering strategy to confine lithium diffusion at interlayer space of MoS₂, thus precisely define the 1T phase transition.

Figure R6 (for review only). Illustration of strain engineering strategy to confine lithium intercalation in 2D layered materials.

References:

1. Chen, L. J. Metal silicides: An integral part of microelectronics. *JOM* **57**, 24–30 (2005).
2. Tang, W. et al. Ultrashort channel silicon nanowire transistors with nickel silicide source/drain contacts. *Nano Lett.* **12**, 3979–3985 (2012).
3. Kappera, R. et al. Phase-engineered low-resistance contacts for ultrathin MoS₂ transistors. *Nat. Mater.* **13**, 1128–34 (2014).

Q7:

7. Related to multivalued memory, in Figure 4d, it is apparent that the memory state drifted upward to high conductance values after 10³ repeated cycles, the authors should explain underlying physics and how this could be improved.

Reply:

We thank the reviewer’s comments. In response to reviewer’s concern, we would like to explain the state drifted through experiment. As shown in following **Figure R7a**, during the endurance test, trap states will be generated in hBN due to the impact ionization process. Charge trapping in these traps will induce threshold shift of the memory (**Figure R7b**), thus the gradual shift of its readout states. Such effect is confirmed in our experiment by stress induced threshold drift of the device. In following **Figure R7c**, by applying a constant bias voltage for different time, we clearly observe the threshold shift of MoS₂ transistor with increasing the stress time, which is consistent with the above explanation.

To relief the mentioned drift of readout states, we note that at present stage, the P/E speed is quite asymmetric, because the erase process requires higher operation voltage. By improving the charge injection process for erasing, or balancing it with program, we expect that the stress effect can be alleviated. However, this would require further understanding of the erase process, and investigation to stress induced charge trapping in hBN layer under high field conditions.

Figure R7 (for review only). The time dependent threshold voltage curve to quantify the drift of states. a, Schematic of the endurance test structure with charge trap. **b**, Trend schematic of the threshold voltage drift with increase negative/positive stress time. **c**, Stress time versus threshold voltage curve. The stress voltage amplitude is fixed in 4 V with opposite polarity, and the stress time is both increased from 0.01 s to 100 s.

Revision

In response to reviewer's concern, we've revised the following discussion in revision at page 11 Line 24-27 in the revised manuscript.

“Though hBN is exfoliated as single crystal, it still exhibits certain level of trap states that could store charges during repeated P/E cycles. Under the high electric field, more traps will be created by the defect generation, which in turn exacerbates the drift of memory states.⁴⁶”

REVIEWERS' COMMENTS

Reviewer #2 (Remarks to the Author):

The author has satisfactorily addressed my comments.

For reviewer 1's comment on novelty, it was in my opinion not fully addressed. The authors claim that edge contact is novel, but elsewhere they write " both contacts coexist in our devices, ...".

Furthermore, the HAADF-STEM image of Fig.1b also shows a small edge contact area, so it cannot be significant.

Reviewer #3 (Remarks to the Author):

The authors have provided substantial data for addressing my concerns. The current version has been largely improved, and in my opinion it is ready for publication.

I have also re-evaluated the reply to reviewer 1's comment. Overall, I think the concerns are properly addressed now.

The biggest concern from reviewer 1 is the novelty, but to my best knowledge the authors present a first study to use the phase-engineered-edge-contact approach to concurrently improve the performance of endurance and operation speed, which has been stated in the author's reply.

For the mechanism concern, the authors provided new evidence that supported their claims, and updated their discussion by including the latest literature on 2D flash memory. Meanwhile, the extracted Schottky barrier under float gate modulation supported that edge contact is more tunable than typical top contact, which is consistent with their CV analysis. Such characteristic is likely to promote charge injection from metal contact and improve charge injection efficiency in the fabricated memory cell.

Point-to-point reply to Reviewers' comments

Reviewer 2

The author has satisfactorily addressed my comments.

Q1.

For reviewer 1's comment on novelty, it was in my opinion not fully addressed. The authors claim that edge contact is novel, but elsewhere they write " both contacts coexist in our devices, ...".

Reply:

We'd like to clarify that in the fabricated edge contacted memory cells, the top contact area was designed to not overlap with the bottom Gr float gate, so that it does not contribute to tunneling current through hBN. As illustrated in Figure R1, in the paired memory cell (Figure 2), the contact to channel is made of 1T-MoS₂ in an edge contact configuration, the additionally defined Cr/Au contact to MoS₂ does not overlap with bottom graphene layer. According to the tunneling pathway shown in figure, the latter contact should not influence the charge injection to float gate due to its long tunneling distance. Thus, in our devices, we've intentionally reduced the direct contact area of Cr/Au contact to 2H-MoS₂ and avoided their overlap with Gr float gate to suppress its influence to memory operation. According to our previous analysis on tunneling current density through conventional Cr/Au contact to 2H MoS₂, which is ~2 orders lower than edge contact, the impact of Cr/Au contact to peripheral 2H MoS₂ around 1T-MoS₂ to the memory operation speed is negligible.

Figure R1. Illustration of the contact configuration in edge contact device shown in Figure 2b.

Revision:

To improve the clarity, we've added the following discussion in main manuscript.

At page 6, line 3-8

"in our case, the measured tunneling current density via 1T-MoS₂/hBN/Gr pathway is ~2 order lower than that through 2H-MoS₂ due to the higher tunneling barrier from the Fermi level of 1T-MoS₂ (Supplementary Note 11). By comparing directly the tunneling current across edge contacted 2H-MoS₂/hBN/graphene heterostructure, we

found ~2 orders enhancement to conventional top contacted one at the electric field of 8 MV/cm (Supplementary Note 11). This suggest that the edge contact interface plays a critical role in enhancing the charge injection efficiency.”

Q2.

Furthermore, the HAADF-STEM image of Fig.1b also shows a small edge contact area, so it cannot be significant.

Reply:

Due to the diffusion of Li in interlayer of MoS₂, the prepared phase change interface in MoS₂ extrudes from the defined PMMA window in our case. The randomness of Li diffusion causes roughness in the defined phase engineered contact interface. As illustrated in Figure R2, the deviation of phase change boundary in the in-plane and out-plane direction (among layers) make it difficult to capture an ideal atomic arrangement at the interface of 1T-2H MoS₂ in the cross-section. In our case, the atomic arrangement in the image shown in Figure 1b apparently deviated from a perfect 2H-structure. However, due to the mixed presence of 1T and 2H phases along the electron beam, the atomic arrangement does not match identically with pure 1T phase. At the bottom layer, we do observe a clear 1T to 2H transition, which we interpret it a result of local clean interface.

Figure R2. Illustration of the contact configuration in the interface of 1T/2H MoS₂ shown in Figure 1b.

We note that in addition to the cross-section image, photoluminescence (PL) quenching of 2H-MoS₂ (Supplementary Figure 7) is a good signature to confirm phase transition into its metallic 1T phase due to the escape depth of luminescence signal >10 nm. In our case, the quench of PL signal is confirmed during the fabrication of each sample to make sure the formation of edge contact to MoS₂ channel.

Reviewer 3.

The authors have provided substantial data for addressing my concerns. The current

version has been largely improved, and in my opinion it is ready for publication.

I have also re-evaluated the reply to reviewer 1's comment. Overall, I think the concerns are properly addressed now.

The biggest concern from reviewer 1 is the novelty, but to my best knowledge the authors present a first study to use the phase-engineered-edge-contact approach to concurrently improve the performance of endurance and operation speed, which has been stated in the author's reply.

For the mechanism concern, the authors provided new evidences that supported their claims, and updated their discussion by including the latest literature on 2D flash memory. Meanwhile, the extracted Schottky barrier under float gate modulation supported that edge contact is more tunable than typical top contact, which is consistent with their CV analysis. Such characteristic is likely to promote charge injection from metal contact, and improve charge injection efficiency in the fabricated memory cell.

Reply:

We sincerely thank the reviewer comments and valuable suggestions that helped us to improve the quality of the manuscript.